# Chromosome separation during *Drosophila* male meiosis I requires separase-mediated cleavage of the homolog conjunction protein UNO

**Joe Weber**, **Zeynep Kabakci**, **Soumya Chaurasia**, **Erich Brunner**, **Christian F. Lehner** *

Department of Molecular Life Science (DMLS), University of Zurich, Zurich, Switzerland

* christian.lehner@imls.uzh.ch

**Data Availability Statement:** All relevant data are within the manuscript and its Supporting Information files.

## Abstract

Regular chromosome segregation during the first meiotic division requires prior pairing of homologous chromosomes into bivalents. During canonical meiosis, linkage between homologous chromosomes is maintained until late metaphase I by chiasmata resulting from meiotic recombination in combination with distal sister chromatid cohesion. Separase-mediated elimination of cohesin from chromosome arms at the end of metaphase I permits terminalization of chiasmata and homolog segregation to opposite spindle poles during anaphase I. Interestingly, separase is also required for bivalent splitting during meiosis I in *Drosophila* males, where homologs are conjoined by an alternative mechanism independent of meiotic recombination and cohesin. Here we report the identification of a novel alternative homolog conjunction protein encoded by the previously uncharacterized gene *univalents only* (*uno*). The univalents that are present in *uno* null mutants at the start of meiosis I, instead of normal bivalents, are segregated randomly. In wild type, UNO protein is detected in dots associated with bivalent chromosomes and most abundantly at the localized pairing site of the sex chromosomes. UNO is cleaved by separase. Expression of a mutant UNO version with a non-functional separase cleavage site restores homolog conjunction in a uno null background. However, separation of bivalents during meiosis I is completely abrogated by this non-cleavable UNO version. Therefore, we propose that homolog separation during *Drosophila* male meiosis I is triggered by separase-mediated cleavage of UNO.

## Author summary

For production of sperm and oocytes, precursor cells have to complete meiosis, which comprises two successive divisions. These divisions convert diploid cells into haploid cells. Diploid cells are characterized by the presence of both a maternal and a paternal copy of each chromosome. In contrast, haploid cells have only one copy of each chromosome. Chromosome copy reduction during meiosis requires the pairing of the maternal and paternal copy of each chromosome into a bivalent chromosome in the diploid

**Funding:** The research was supported by funds obtained from the Swiss National Science Foundation (www.snf.ch), grant number 31003A_179433 (CFL) and from the Fonds National de la Recherche Luxembourg (www.fnr.lu), fellowship number AFR_PhD 11273596 (JW). The funders had no role in study design, data collection and analysis, decision to publish, or preparation of the manuscript.

**Competing interests:** The authors have declared that no competing interests exist.

precursor cell. Bivalents are integrated into the spindle in a bi-polar fashion, with the two chromosomes connected to opposite spindle poles. Once all bivalents are properly bi-oriented, the linkage between the two chromosome copies is destroyed so that they can be pulled apart towards opposite poles. While all sexually reproducing species have to accomplish meiosis, the mechanism for linkage of the two chromosome copies into a bivalent is not invariant. A special type of linkage is used in males of the fly *Drosophila melanogaster* for example. Here we describe UNO, a novel protein required for this chromosome linkage. When the linkage needs to be destroyed for chromosome separation during the first meiotic division, UNO is cleaved by a protease known as separase.

## Introduction

The partitioning of genetic information during both mitotic and meiotic divisions is governed by a common principle. Pairs of chromosomal entities are first bi-oriented within a bipolar spindle. For bipolar orientation, each of the chromosomal entities has to be equipped with one kinetochore that allows attachment to spindle microtubules. Moreover, a linkage between the paired chromosomal entities needs to be stable until they are all bi-oriented. This linkage is then destroyed towards the end of metaphase. As a result, the spindle can pull the separated chromosomal entities apart towards opposite spindle poles during anaphase.

Pair production before mitotic divisions occurs during S phase when chromosomal DNA is replicated into two identical sister chromatids. DNA replication is coupled with linkage of sister chromatids by cohesin complexes [1–5]. These complexes consist of a heterodimer of SMC proteins, SMC1 and SMC3, which associate with additional subunits [1,2]. One of these, an α kleisin protein, is of particular importance for the timely resolution of sister chromatid cohesion before anaphase onset. The α kleisins have N- and C-terminal domains that bind to SMC3 and SMC1, respectively [6–8]. Moreover, within the intervening linker region, α kleisins have motifs that are recognized and cleaved by separase [9–11]. This endoprotease is activated just before anaphase onset after proteasomal degradation of securin, a separase inhibitory protein [12–17]. Securin proteolysis is triggered by polyubiquitination mediated by the ubiquitin ligase designated as anaphase promoting complex/cyclosome (APC/C) [18,19]. As a result of separase-mediated α kleisin-cleavage, cohesion between sister chromatids is resolved, allowing their segregation to opposite spindle poles [1].

In comparison to mitotic divisions, the proper pairing of two appropriate chromosomal entities before the first meiotic division is more demanding. In contrast to sister chromatids that arise in close spatial vicinity from DNA replication, the homologous chromosomes that need to be paired into bivalents before meiosis I (M I) are not necessarily in close spatial vicinity within the cell nucleus. The molecular mechanisms that promote the correct pairing of homologous chromosomes free of entangling with non-homologous chromosomes during M I are still poorly understood.

Several contributing processes have been described and their significance can vary between different organisms [20]. The last mitotic division before M I arranges chromosomes into a Rabl configuration, where centromeres are clustered on the nuclear side close to the spindle pole and telomeres towards the opposite side. In several species, centromere clustering has been implicated in promoting homolog pairing [21]. Similarly, telomere anchoring at the nuclear envelope and subsequent clustering and/or rapid prophase movements are involved in the pairing process in various organisms [22,23]. In addition, meiotic recombination is often of paramount importance for homolog pairing [20].

Recombination is initiated after the pre-meiotic S phase when programmed double strand breaks (DSBs) are induced by dedicated Spo11 orthologs [20,24]. Single strands resulting after DSB processing can invade homologous double stranded DNA by strand exchange, offering a precise mechanism for pairing specifically homologous chromosomes. However, in some species like *Caenorhabditis elegans* and *Drosophila melanogaster*, homolog pairing does not depend on meiotic recombination [20,25]. The pairing of homologous chromosomes usually culminates in the formation of the synaptonemal complex (SC), a highly ordered structure that links homologous chromosomes eventually all along their length [20,25,26]. During canonical meiosis, the SC is a transient structure. But homolog pairing is maintained beyond SC disassembly by cross-overs (COs) that result from repair of some of the meiotic DSBs by recombination between homologous non-sister chromatids. COs in combination with sister chromatid cohesion, which precludes terminalization of the COs, maintain homologous chromosomes paired until the metaphase-to-anaphase transition of M I. As in mitosis, it is the separase-mediated cleavage of the α kleisin subunit of cohesin complexes that allows the bipolar segregation of homologous chromosomes during M I [27–29]. After resolution of sister chromatid cohesion, the pulling forces in the spindle move the homologous chromosomes apart during anaphase I and COs are terminalized. Importantly, in contrast to mitosis, α kleisin cleavage during M I occurs in a regionally controlled manner exclusively along the chromosome arms and not within the pericentromeric region. The pericentromeric sister chromatid cohesion which perdures beyond M I maintains sister centromeres paired, allowing their bipolar integration into the M II spindle. After separase mediated destruction of pericentromeric sister chromatid cohesion during late metaphase II, sister centromeres are moved apart during anaphase II. The crucial, regionally controlled cleavage of α kleisin during the meiotic divisions depends on the expression of meiosis-specific Rec8 α kleisins [29–32].

Among the numerous variants of meiosis that have evolved, the one observed in males of higher dipterans including *D. melanogaster* is particularly striking. In these dipteran species, SC formation and meiotic recombination do not occur during male meiosis, while these processes are clearly essential for meiosis in the female sex. Despite absence of SC and COs, bivalents are also formed in *Drosophila* spermatocytes before M I, indicating the function of an alternative homolog conjunction (AHC) system. Genetic screens have led to initial insight into the molecular basis of AHC. Three genes (*mnm*, *snm* and *tef*) were identified as being specifically required for AHC [33–35]. Mutations in these genes compromise chromosome segregation during male but not female meiosis. In mutant spermatocytes, bivalents fall apart prematurely into univalents, which segregate randomly during M I. While all four bivalents are affected in *mnm* and *snm* mutant spermatocytes [35], only the three autosomal bivalents (of chromosome 2, 3 and 4) are perturbed in *tef* mutants [33]. The two sex chromosomes, Y and X (chromosome 1), are strongly heteromorphic in *Drosophila*. The degeneration of the Y chromosome during evolution and consequential loss of homologous sequences for canonical pairing with the X chromosome has presumably led to an alternative pairing system for the sex chromosomes, which might have served as a pre-adaptation for the complete loss of canonical pairing during male meiosis. The pairing of the heteromorphic sex chromosomes in *Drosophila* spermatocytes relies on the rDNA repeat clusters that are present on both X and Y chromosome (chr). More specifically, the 240 bp repeats within the intergenic spacers (IGS) of the rDNA units were shown to be required and sufficient for X-Y pairing [36–38]. In contrast to sex chromosome pairing, autosomal pairing does not appear to involve comparable specifically localized pairing centers [39–41].

The molecular characterization of the *mnm* and *snm* genes [35] has provided strong evidence that the corresponding protein products are directly involved in the linking of chromosomes into bivalents. The gene *mnm* (abbreviated for *modifier of mdg4 in meiosis*) is part of a

complex genetic locus, *mod(mdg4)*, which generates over 30 alternatively spliced transcripts, often involving trans-splicing [42–44]. The different transcripts code for proteins with a common N-terminal BTB/POZ domain and a distinct C-terminal part that usually includes a unique FLYWCH Zn-finger domain. Those *mod(mdg4)* proteins that have been studied function primarily in chromatin organization, but they are not thought to bind DNA directly. The BTB/POZ domain mediates homo- or heterodimerization and the FLYWCH Zn-finger domains have also been implicated in protein interaction [45]. The gene *snm* (abbreviated for *stromalin in meiosis*, FlyBase designation *SA-2*) is a distant relative of the stromalins (SCC3/SA/STAG protein family) [35]. While stromalin proteins bind to α kleisin and function as subunits within cohesin complexes [1,46,47], SNM is clearly not co-localized with cohesin core subunits and therefore does not function as a cohesin complex subunit [35]. SNM and also MNM protein accumulate during spermatocyte maturation primarily within subnucleolar foci that overlap with many of the fluorescence *in situ* hybridization (FISH) signals obtained with an IGS 240 bp repeat probe [35,38]. At the start of the first meiotic division, these subnucleolar foci coalesce into a single very prominent dot on the sex chromosome bivalent [35]. In addition, far weaker MNM spots are detectable on the autosomal bivalents [35]. Intriguingly, MNM and SNM disappear rapidly from chromosomes during the metaphase to anaphase transition of M I [35] in a separase-dependent manner [48], and they are no longer detectable during M II.

The role of the protein product expressed from the gene *tef* (abbreviated for *teflon*) is more enigmatic compared to MNM and SNM. The TEF protein includes three C2H2-type zinc fingers, one within the N- and two within the C-terminal region [34]. So far, localization of TEF within spermatocytes has not been successful, even with a *tef-GFP* transgene that is functional based on mutant rescue experiments [34].

Evidently, many crucial questions concerning AHC in *Drosophila* spermatocytes remain unanswered. While MNM and SNM are very likely to function directly in a kind of chromosome glue, it remains to be clarified how such a glue exclusively links homologous chromosomes. MNM and SNM do not appear to have DNA binding domains. While TEF has such domains, it is not required for MNM/SNM localization to the sex chromosome pairing center and for sex chromosome conjunction, and it might not be present on bivalents at the onset of the first meiotic division. Moreover, it is unknown how the AHC is resolved at the metaphase-to-anaphase transition of M I. While separase is known to be required, its substrate remains to be identified. The three proteins, MNM, SNM and TEF, do not appear to be separase substrates according to our preliminary analyses, suggesting an involvement of additional proteins in AHC.

For an identification of additional proteins involved in AHC, we have initiated a proteomic approach. A procedure for mass isolation of thousands of testes from *Drosophila* late larvae and early pupae was developed, overcoming traditional sample limitation with hand dissection. This allowed the identification of a previously uncharacterized protein co-purifying with EGFP fusions of MNM and SNM. As described here, the function of the corresponding gene, *univalents only* (*uno*), is essential for AHC. Moreover, UNO protein cleavage by separase permits homolog separation during M I.

## Results

### UNO is found in association with the alternative homolog conjunction proteins MNM and SNM

To search for homolog conjunction proteins that might function in association with MNM and SNM, we sought to perform affinity-isolation of these AHC proteins followed by

identification of co-purifying proteins by mass spectrometry. To obtain sufficient material, we developed a mass isolation procedure for obtaining thousands of testes from late larvae and early pupae (Fig 1A). The testes were isolated from animals expressing EGFP fusion proteins from *UASt* transgenes driven by a *bamP-GAL4-VP16* transgene (*bam>mnm-EGFP* and *bam>snm-EGFP*, respectively). In *bam>mnm-EGFP* and *bam>snm-EGFP* testes, the EGFP fusion proteins are present in spermatocytes throughout their maturation (stages S1 –M3) [49] and they are known to be functional [50]. For the identification of irrelevant proteins, co-purifying because of association with EGFP or anti-EGFP beads, control testes were analyzed in parallel. These control testes expressed an *exumP-EGFP* transgene under control of a fragment from the *exu* cis-regulatory region. Expression of *exumP-EGFP* occurred exclusively in spermatocytes, similar as the *bamP-GAL4-VP16* driven expression of *mnm-EGFP* and *snm-EGFP*, although a bit later (starting at the S3 rather than the S1 stage). The fluorescent EGFP signals present in all the three genotypes were exploited for a testis purification step with a BioSorter, a commercially available device for large particle sorting and analysis. After sorting, testes were homogenized. Homogenates were used for affinity-purification with anti-EGFP beads and co-purifying proteins were characterized by MS/MS analyses. Four replicate experiments were performed with each of the three genotypes.

We identified a total of 903 proteins (Fig 1B; S1 Table), three of which were specifically associated with MNM-EGFP and 72 specifically with SNM-EGFP. Hundred and twenty-three were co-purified with both MNM-EGFP and SNM-EGFP, but not with EGFP (Fig 1B). Gratifyingly, endogenous SNM and TEF were among the proteins co-purified with MNM-EGFP. Moreover, endogenous MNM was detected among the proteins co-purified with SNM-EGFP (Fig 1C). None of the known AHC proteins (MNM, SNM, TEF) were detected in the four control EGFP purifications to a comparable extent (Fig 1C). These results provide the first evidence that the known AHC proteins form protein complexes, consistent with the previously described co-localization of MNM and SNM in particular during the first meiotic division [35].

The predicted *CG8712* product was among the proteins that were co-purified very effectively with both MNM-EGFP and SNM-EGFP, but not EGFP (Fig 1C, S1 Fig). Moreover, *CG8712* transcript levels are maximal in testis [51,52]. Therefore, we started a detailed characterization of this gene (Fig 1D). In the absence of *CG8712* function (see below), univalents instead of bivalents were observed at the start of the first meiotic division in spermatocytes. Accordingly, *CG8712* was named <u>univalents only</u> (*uno*).

Our searches for *uno* orthologs detected these exclusively in the genomes of higher dipterans (including drosophilid, tephritid, muscid and calliphorid species) (S2 Fig). In these species, chiasmata are thought to be absent during male meiosis [53]. In contrast, we failed to detect *uno* orthologs in lower dipterans (like *Annopheles*, *Aedes* and *Culex*) where chiasmata during male meiosis have been reported [53]. Moreover, we also failed to detect *uno* related sequences in the genome sequences of more distantly related species. Therefore, the taxonomic distribution of *uno* genes appears to be identical to that of *snm* and *tef* [34,35]. Overall, the results of our bioinformatic analyses suggest that UNO, like SNM and TEF, is restricted to dipteran species that rely on AHC instead of canonical homolog pairing during male meiosis. In case of MNM, comparable analyses were not attempted, because it is encoded by the highly complex *mod(mdg4)* locus. While this locus appears to have been present already before the divergence of the insect lineages [54], it remains a possibility that the terminal *mnm*-specific exon might have arisen much later in the progenitor of the higher dipterans during the same time frame as the other AHC genes.

The predicted UNO amino acid sequence did not include any known protein domains. However, the comparison of drosophilid UNO sequences revealed regions of higher

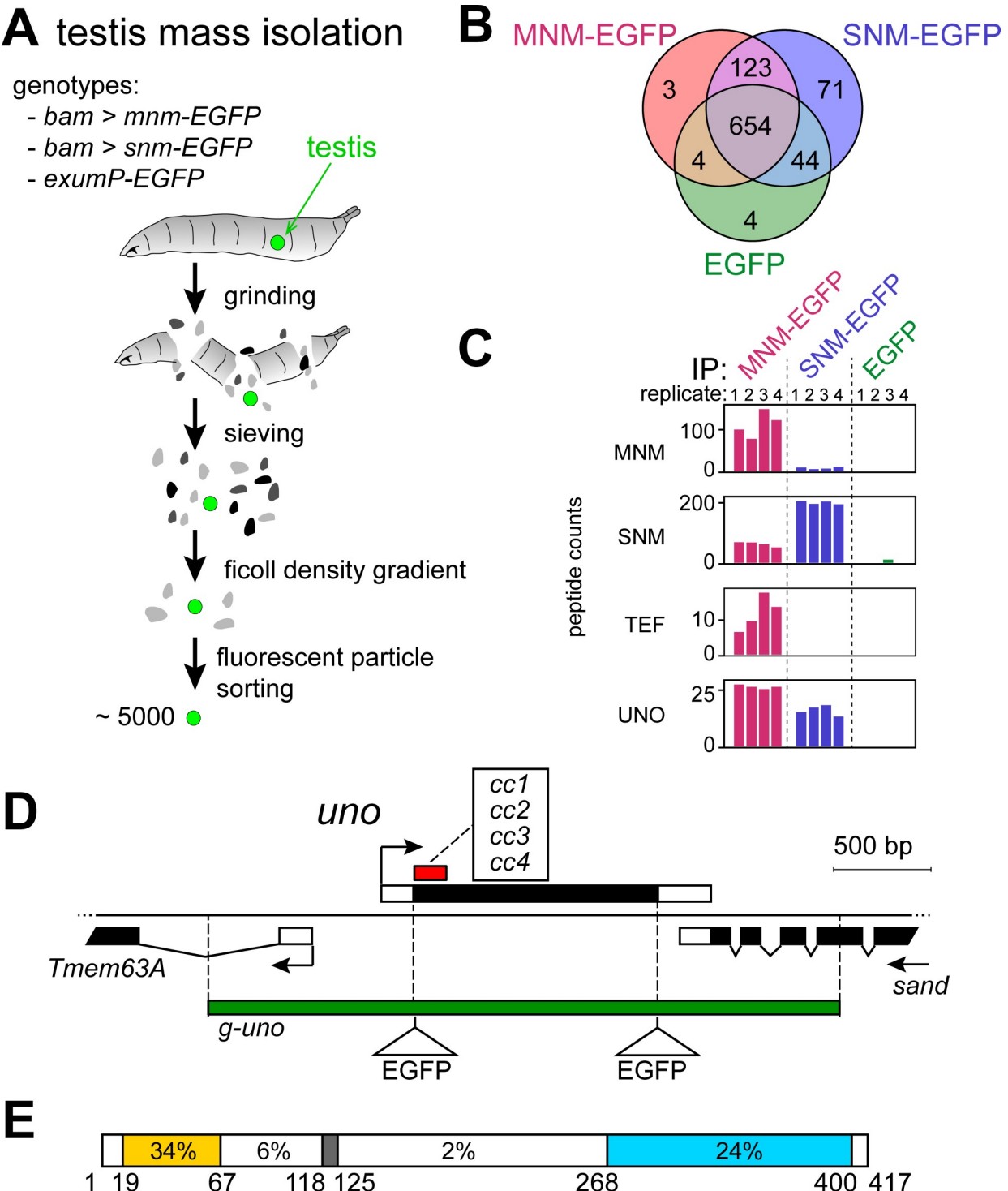

**Fig 1. UNO is co-purified with MNM-EGFP and SNM-EGFP. (A)** Testis mass isolation. After grinding up third instar wandering stage larvae and early pupae with EGFP expression in testes, sieving and density gradient centrifugation were used for enrichment of testes. Fluorescent particle sorting was applied as the final purification step. The procedure allowed an isolation of around five thousand testes per run. For each of the three genotypes (bam>mnm-EGFP, bam>snm-EGFP, and exuM-EGFP) around 30'000 testes were prepared for four replicates of affinity-purification with anti-EGFP beads. **(B)** VENN diagram summarizing the numbers of co-purified proteins identified by mass spectrometry after affinity-purification of MNM-EGFP, SNM-EGFP or EGFP from testes extracts. **(C)** Bar diagram presenting the spectral counts of peptides derived from MNM, SNM, TEF and UNO that were detected in each of the four replicate samples obtained by affinity-purification of MNM-EGFP, SNM-EGFP and EGFP from testis extracts. **(D)** The function of *univalents only* (*uno*), a gene coding for a protein co-purified with

MNM-EGFP and SNM-EGFP, was further characterized. Four distinct, intragenic out-of-frame deletions (cc1 –cc4) generated by CRISPR/cas eliminate parts of the coding region (black box) as indicated approximately by the red box. The green bar indicates the genomic region present in the *g-uno* transgenes, which had either an N-terminal, a C-terminal, or no in-frame fusion to the EGFP coding sequence. **(E)** Amino acid sequence conservation in UNO. *Drosophila melanogaster* UNO contains 417 amino acids. Positions and extent of amino acid identity observed in different regions based on an alignment of drosophilid UNO orthologs (S2 Fig) are indicated. Two regions with stronger sequence conservation close to the N- and C-termini (yellow and blue, respectively) are connected by a poorly conserved linker region, which includes a well-conserved match to the separase cleavage site consensus (grey).

conservation close to the N- and C-termini (Fig 1E, S2 Fig). These more conserved UNO regions are connected by a highly variable linker region.

Interestingly, UNO was found to have similarity with the predicted protein product of another uncharacterized *Drosophila* gene, *CG32117* (S3 Fig). The *CG32117* protein consists of a tandem repeat of an amino acid sequence that shares extensive similarity with the conserved N-terminal region of UNO (S3 Fig). Putative *CG32117* orthologs have a taxonomic distribution that is more restricted than that of *uno*; they were detected in drosophilid species, but not in tephritids, muscids and calliphorids (S3 Fig). *CG32117* might thus be an evolutionary descendant of *uno*.

## UNO is required for conjunction and regular segregation of homologs during male meiosis I

The co-purification of UNO with MNM-EGFP and SNM-EGFP suggested that this novel protein might also function in AHC. For an initial evaluation of this notion, we performed spermatocyte-specific UNO depletion by transgenic RNAi and analyzed the consequences with testis squash preparations.

During wild-type spermatogenesis (Fig 2A) [55], the germline stem cells, hosted in the niches at the closed distal ends of the two testis tubes present in a male, divide asymmetrically. The differentiating daughter cell gets enveloped by two somatic cyst cells. Within such cysts, the differentiating germline cells progress through four transit-amplifying cell division cycles with incomplete cytokinesis, generating a cluster of 16 interconnected spermatocytes. The spermatocytes grow and mature during progression through the stages S1—S6 [49]. Pre-meiotic S phase and chromosome pairing into bivalents are completed rapidly during the S1/S2a stages. Thereafter the process of chromosome territory formation breaks up non-homologous chromosome associations during the S2b/S3 stages. Chromosome territories arise within the spermatocyte nucleus because the three large bivalents, those of the sex chromosomes (chr XY) and the two large autosomes (chr 2 and chr 3), are separated from each other. The bivalent of the small dot-like autosome (chr 4) usually remains close to or within the sex chromosome territory. Chromosome condensation in preparation for the first meiotic division starts during the S6 stage and converts the territories into compact chromosome clumps. The subsequent two meiotic divisions generate cysts with 64 haploid round and connected spermatid cells, which eventually differentiate into highly elongated sperm that are individualized and released at the proximal end of the testis tube into the seminal vesicle.

After spermatocyte-specific UNO depletion by transgenic RNAi, the early stages (S1—S3) always displayed a normal pattern of DNA staining. At the S3 stage, the three major chromosome territories were readily apparent as in controls (Fig 2B). However, at the late spermatocyte stages (S5 and S6) the appearance of the territories was usually abnormal. In comparison to control spermatocytes, individual chromosome territories were more bi-lobed or even separated into two sub-territories (Fig 2B). Chromosome condensation during S6 and early meiosis I exposed this territory subdivision even more strongly. During prometaphase I, the number of chromosome clumps was almost invariably increased above the normal number of

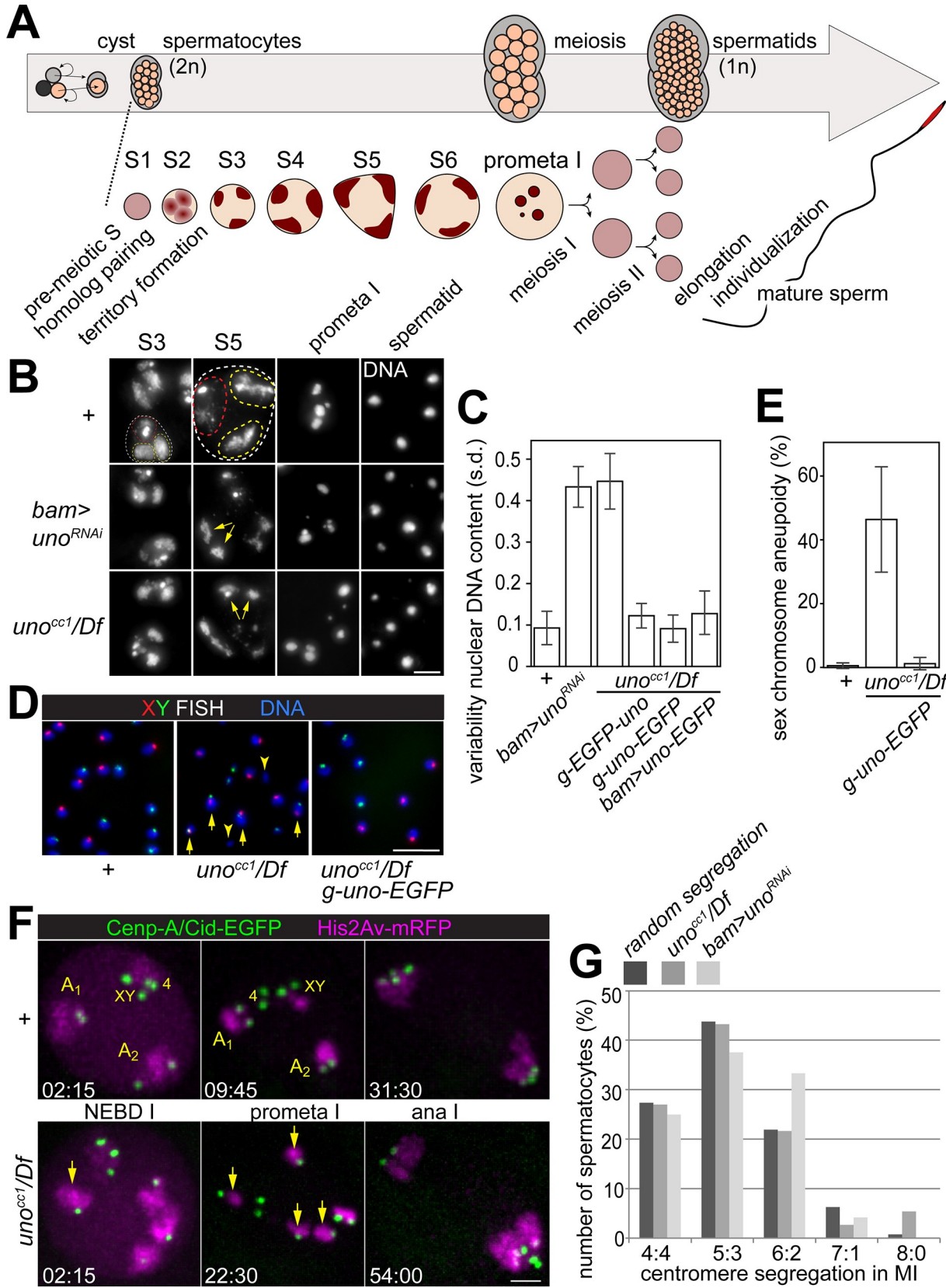

**Fig 2. UNO is required for conjunction and regular segregation of homologs during male meiosis I. (A)** Spermatogenesis in *Drosophila melanogaster*. See text for further explanations. **(B)** Squash preparations of the testes dissected from the indicated genotypes *(+, or bam>uno*^RNAi^, or *uno*^cc1^*/ Df(2R)Exel7094)* were labeled with a DNA stain. Representative images illustrate the subnuclear pattern of DNA staining at selected stages (spermatocytes at the S3 and S5 stages, in prometaphase I, and early postmeiotic spermatids). A nucleus of a spermatocyte at the S3 stage in control (dashed line) contains usually already three distinct major territories of DNA staining. In the far larger nucleus at the S5 stage, these territories are also displayed in the control, containing either the sex chromosome bivalent (red dashed line) or the two large autosomal bivalents (chr 2 and chr 3) (yellow dashed lines). In genotypes with compromised *uno* function, autosomal territories have often a split appearance at the S5 stage (yellow arrows), while a normal appearance is observed at the S3 stage. The characteristic number of four bivalents (three large and one small) are present during prometaphase I in the control, while an increased number of chromosomes is detected in the other genotypes. Moreover, the nuclei of early spermatids are homogenous in size and DNA content in control, but highly variable in the other genotypes. **(C)** Quantification of the variability of the DNA content in early postmeiotic spermatid nuclei. Squash preparations of testes from different genotypes *(+, or bam>uno*^RNAi^, or *uno*^cc1^*/ Df(2R)Exel7094* without and with the indicated additional transgenes) were labeled with a DNA stain. Cysts with early postmeiotic spermatids were imaged, DNA staining intensity in each nucleus was quantified, and the standard deviation as a measure of DNA content variability among the nuclei within a given cyst was determined. The bar diagram displays the average (+/- s.d.) of the standard deviations obtained for each of the spermatid cysts analyzed for a given genotype (n > 11 cysts from multiple males). **(D,E)** FISH with a red fluorescent probe for chr X and a green fluorescent probe for chr Y was used to analyze the rate of sex chromosome missegregation in the indicated genotypes. **(D)** Normal segregation results in either a red or a green signal per spermatid nucleus, as in the left- and right most panels. In contrast, missegregation results in spermatid nuclei of variable sizes without any or with both red and green signals, as in the middle panel. **(E)** Bar diagram displaying the average percentage of nuclei with abnormal XY FISH signals obtained after analysis of multiple cysts for a given genotype. Standard deviations are shown as well. The number of analyzed early spermatid cysts was 11 for *w*^1118^ (+) and 13 for each of the other two genotypes (*uno*^cc1^*/ Df(2R)Exel7094* either without or with *g-uno-EGFP*). **(F,G)** Time lapse imaging of chromosome segregation during M I in control (+) and in *uno* mutants (*uno*^cc1^*/ Df(2R)Exel7094*) expressing Cenp-A/Cid-EGFP and His2Av-mRFP. **(F)** Representative still frames at the stage of NEBD I, prometaphase I and anaphase I. In control, chromosome condensation during NEBD I converts the chromosome territories of the two large autosomes (A1, A2) and of the sex chromosome (XY) with associated chr 4 (4) into compact bivalents, followed by 4:4 segregation of centromeric Cenp-A/Cid-EGFP dots during anaphase I. In *uno* mutants, univalents (i.e., smaller masses of His2Av-mRFP with only one associated Cenp-A/Cid-EGFP dot; examples indicated by yellow arrows) are detected during prometaphase I. Moreover, a 2:6 segregation is evident in the displayed *uno* mutant anaphase I. Time (minutes:seconds) after the last time point before onset of NEBD I is indicated. **(G)** Histogram describing the relative frequency of the five possible centromere segregation patterns (4:4, 5:3, 6:2, 7:1 and 8:0) as observed by time lapse analysis of M I with spermatocytes expressing Cenp-A/Cid-EGFP and His2Av-mRFP in the indicated genotypes with compromised *uno* function. Moreover, the relative frequencies theoretically predicted for the case of an entirely random segregation of the eight centromeres is given as well (random segregation). n = 37 and 24 spermatocytes from *uno* mutant and *uno* depleted testes, respectively. Scale bars = 5 μm (B), 10 μm (D) and 3 μm (F).

three large and one small clump. Moreover, the size of the chromosome clumps was reduced (Fig 2B). Close to eight such chromosome clumps were usually observed in mutant prometaphase I cells, consistent with a presence of univalent instead of bivalent chromosomes. In postmeiotic cysts, size and DNA content of the spermatid nuclei was always far less homogenous compared to controls (Fig 2B, 2C, S4 Fig). This highly variable appearance of spermatid nuclei most likely reflects chromosome missegregation during the meiotic divisions. In summary, all the abnormalities observed after spermatocyte-specific UNO depletion appeared identical to those described in *mnm* and *snm* null mutants [35], suggesting that *uno* is indeed required for AHC.

For further confirmation that *uno* is essential and specifically required for AHC, we generated *uno* null alleles using CRISPR/cas in combination with sgRNA pairs designed to delete conserved parts of the *uno* coding region (Fig 1D). In total, we established four distinct alleles (named cc1 –cc4) with out-of-frame deletions within the coding *region*. These deletions started after the first few *uno* codons (after codon 2, 5, 7 and 10 in cc1, cc2, cc3 and cc4, respectively). The four mutant alleles are therefore predicted to express severely truncated proteins comprising at most the first ten N-terminal amino acids of UNO. Zygotes homozygous for these *uno* null alleles, or hemizygous (*uno* null alleles over *Df(2R)Exel7094*), developed to the adult stage. Fertility of *uno*^cc1^*/ Df(2R)Exel7094* was found to be normal in case of adult females, but male fertility was reduced to about 36% of control males. As shown for *uno*^cc1^*/ Df(2R)Exel7094* (Fig 2B), DNA staining revealed the same characteristic abnormalities in testis squash preparations as after spermatocyte-specific *uno* depletion by RNAi (Fig 2B), and as described in *mnm* and *snm* mutants [35].

To address whether chromosomes segregate randomly during M I in *uno* null mutants, which is expected if homologs are not conjoined into bivalents at the start of the first meiotic

division, we performed FISH with a red fluorescent probe for chr X and a green fluorescent probe for chr Y (Fig 2D). After regular segregation of the sex chromosomes during M I, spermatid nuclei are expected to have either a red or a green signal. In contrast, random segregation will also generate nuclei with either no signal at all, or with both a red and a green signal, indicating sex chromosomal aneuploidy. The ratio of nuclei with normal signals (1 green or 1 red) and abnormal signals (0 or both 1 green and 1 red) is expected to be one in case of random segregation during M I. In principle, missegregation during M II can also generate nuclei without a signal, as well as nuclei with two signals of the same color. However, scoring of these latter nuclei was not attempted, as it is unreliable. The two signals of the same color are not necessarily separated sufficiently for unequivocal resolution. Within the given limitations, the fraction of nuclei with abnormal XY FISH signals observed in *uno*^cc1 hemizygous spermatids suggested that the sex chromosomes indeed segregate randomly during M I in the absence of *uno* function (Fig 2E).

For a further clarification of meiotic chromosome missegregation caused by loss of *uno* function, we performed time-lapse imaging using spermatocytes expressing histone H2Av-mRFP (His2Av-mRFP) and Cenp-A/Cid-EGFP for labeling of chromosomes and centromeres, respectively. Analogous analyses in wild-type and *mnm* null mutant spermatocytes have been described recently [56]. Before nuclear envelope breakdown (NEBD), His2Av-mRFP reveals the three major chromosome territories in primary spermatocytes in wild-type M I [56]. Two of these territories represent the two large autosomes and the third territory the sex chromosome bivalent and the usually associated bivalent of the dot-like chr 4. The rapidly accelerating chromosome condensation during NEBD converts the territories into compact blobs. While the His2Av-mRFP marker clearly reveals the large bivalents (chr XY, chr 2 and chr 3), chr 4 contains so little His2Av-mRFP that it is only revealed by the associated centromeric Cenp-A/Cid-EGFP signals (Fig 2F) [56]. Rapid saltatory movements during prometaphase I precede the eventual bi-polar integration of bivalents into a compact metaphase I plate that remains stable for 15–20 minutes until bivalents split in anaphase I (Fig 2F) [56].

In spermatocytes lacking *uno* function (Fig 2F, S1 Movie), bivalents were observed to separate prematurely into univalents. While some bivalents were still intact at NEBD I in *uno* mutants, these were all very rapidly converted into univalents as soon as spindle forces started to act on kinetochores during prometaphase I. While normal spermatocytes contain four bivalents with two centromeres each [56], eight chromosome masses each with a single centromere were observed to display independent saltatory movements in the absence of *uno* function during prometaphase I (Fig 2F, S1 Movie). After a temporally extended phase with these saltatory chromosome movements, most univalents eventually reached stable positions preferentially near the poles, followed by anaphase onset and exit from M I (Fig 2F, S1 Movie). In the absence of *uno* function, the duration from onset of NEBD I until chromosome decondensation in telophase I was extended by about 17 minutes compared to control (in *uno*^cc1/ *Df(2R)Exel7094*: 53 +/- 5 min s.d., n = 8; in *bam>uno*^RNAi: 51 +/- 3 min s.d., n = 28; compared to controls: 35 +/- 2 min s.d., n = 25, [56]). Centromere tracking clearly established that the eight independent centromeres were segregated randomly onto two daughter nuclei during exit from M I (Fig 2G).

During M II in *uno* mutants, time-lapse imaging did not reveal an abnormal chromosome behavior. Normal bi-orientation of sister centromeres into a well aligned metaphase plate, followed by regular segregation during anaphase II was observed (S5 Fig). Without exception, the number of Cid-EGFP signals moving apart to opposite spindle poles during anaphase II was always identical in the two half spindles (n = 37 secondary spermatocytes from four independent cysts) and the two daughter nuclei forming in telophase II were of similar size based on the His2Av-mRFP signals. Moreover, we point out that the results of our XY FISH analysis of *uno* mutant spermatid nuclei provides further support that sister centromeres do not segregate

randomly during M II. In this regard, the ratio ("0/RG") between the numbers of spermatid nuclei without any FISH signals and of those with both red and green signals is informative. 0/RG can be determined reliably. 0/RG is expected to be 1 if sex chromosomes segregate randomly during M I followed by regular sister segregation during M II. In contrast, if random segregation of univalents during M I is followed by random sister segregation during M II, 0/RG is predicted to be 2.77. The observed 0/RG was 0.95 +/- 0.27 (s.d.) on average (n = 13 cysts), indicating that the sisters of sex chromosomes do not segregate randomly during M II.

In summary, analysis of fixed samples, as well as time-lapse imaging of progression through the meiotic division, revealed the same defects in both *uno* null mutants and after spermatocyte-specific *uno* knock-down. Moreover, these abnormalities appeared to be indistinguishable from those described previously in case of *snm* and *mnm* mutants [35,56]. We conclude that *uno* is specifically required for alternative homolog conjunction and regular segregation of homologs during the first meiotic division in males.

To explore whether the *uno*-related gene *CG32117* might also function during meiosis in *Drosophila* males, we generated null alleles using CRISPR/cas. Four *CG32117* alleles (cc1 –cc4) with large intragenic out-of-frame deletions were established (S6 Fig). The predicted mutant protein products include only the first 18–21 amino acids residues followed by premature stops after some extra amino acids. In the hemizygous state (cc1 –cc4 over the deficiencies *Df(3L)ED4486* or *Df(3L)BSC413*, which both delete *CG32117*), the null mutations did not preclude development to the adult stage. The hemizygous males appeared to have a slightly reduced fertility (average 83% of control male fertility). However, we did not detect obvious cytological abnormalities in testis preparations of males lacking *CG32117* function. We conclude that *CG32117* is not essential for viability and fertility under laboratory conditions.

## UNO localizes to meiotic chromosomes until meiosis I

According to transcript profiling data [51,52], *uno* mRNA is present at high levels in testis, at moderate levels in ovaries and at low levels in some other tissues. To characterize expression pattern and subcellular localization of the protein product, we generated antibodies against UNO and used them for immunolabeling of testis samples. To identify specific signals, we stained testes dissected from both control and *uno*[cc1] hemizygous mutant males which cannot express the UNO region used for immunization. Specific anti-UNO signals were observed in spermatocytes from S1 to S6 and during early M I (Fig 3A and 3B). During earlier stages, germline cells appeared to display only background signals, and UNO was not detected in any of the various somatic cell types in testis (Fig 3A). The specific signals in spermatocytes were maximal within the nucleolus (Fig 3B) but also present at lower levels throughout the nucleus (Fig 3A and 3B). The nucleolar signals were rather uniform in early spermatocytes and became increasingly enriched in subnucleolar foci that were evident at the S5 stage (Fig 3B). During the initial stages of the first meiotic division, anti-UNO staining resulted in a very prominent dot that was associated with one of the chromosome bivalents (Fig 3B), the sex chromosome bivalent (see below). In contrast, we did not observe specific anti-UNO signals in early postmeiotic spermatids (Fig 3B) and during later stages.

For independent confirmation of the UNO expression pattern and localization, we established transgenic lines, in which *uno* cis-regulatory sequences control the expression of UNO fused to EGFP at either N- or C terminus (*g-EGFP-uno* and *g-uno-EGFP*) (Fig 1D). To establish transgene functionality, we crossed *g-uno-EGFP* into a *uno* null mutant background (*uno*[cc1]/ *Df(2R)Exel7094*) and analyzed cytologically whether the transgene restored normal sex chromosome segregation during male meiosis and a uniform DNA content in postmeiotic spermatid nuclei. This was indeed the case (Fig 2C, 2E and S4 Fig), indicating that the *g-uno-*

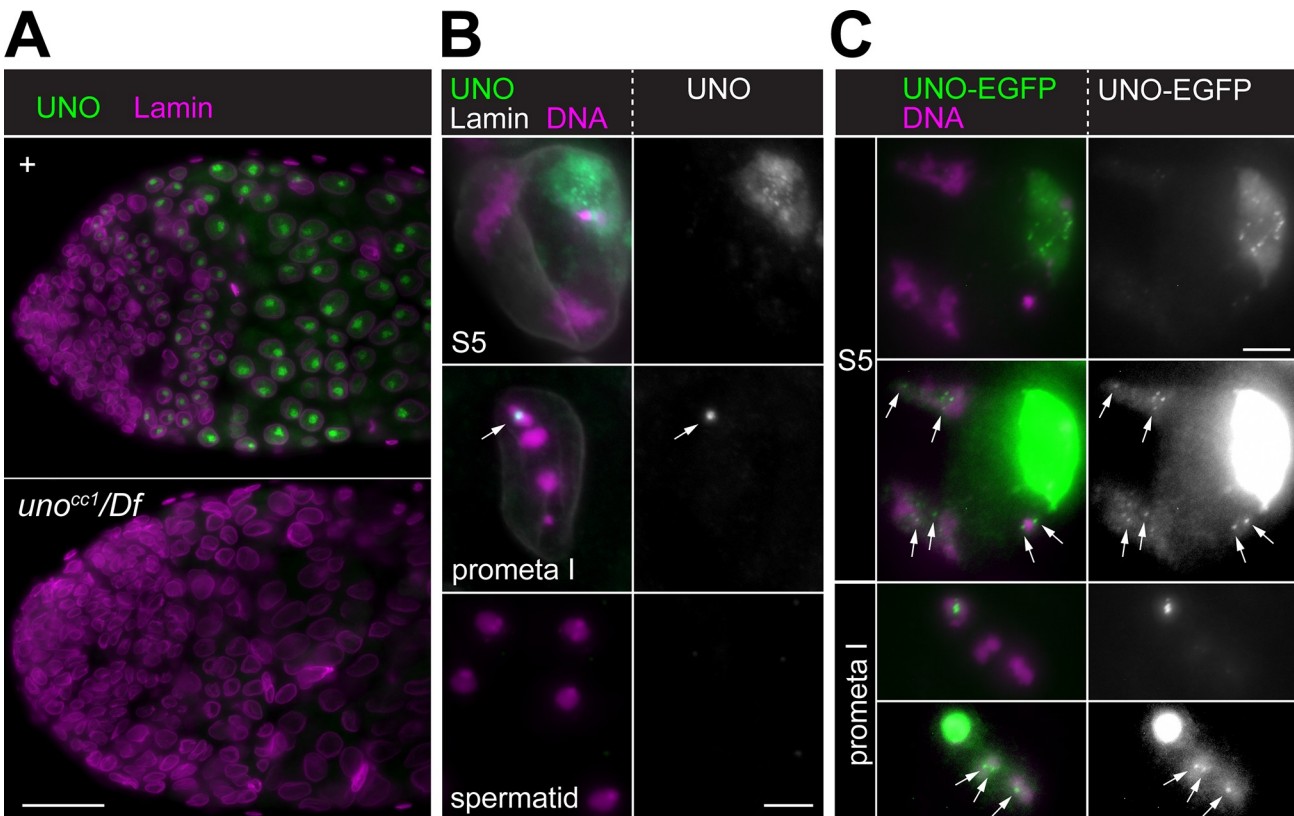

**Fig 3. UNO is localized in chromosome-associated dots until meiosis I. (A,B)** Immunolabeling with anti-UNO. **(A)** Whole mount preparations of testes dissected from either control (+) or *uno*^cc1^ hemizygous males (*uno*^cc1^/*Df*) were labeled with anti-UNO and anti-Lamin. **(B)** Squash preparations of control testes were labeled with anti-UNO, anti-Lamin and a DNA stain. High magnification views of cells at the indicated stages are shown. In S5 spermatocytes (S5), anti-UNO signals are highest in several subnucleolar foci. In prometaphase I (prometa I), after chromosome condensation, these appear to have coalesced into a single bright dot on the sex chromosome bivalent (arrow). Postmeiotic spermatids (spermatid) do not display any specific anti-UNO signals. **(C)** UNO-EGFP signals. Squash preparations were made with testes dissected from *uno* null mutants expressing UNO-EGFP from a transgene under control of the *uno* cis-regulatory region (*uno*^cc1^/*Df(2R)Exel7094*; *g-uno-EGFP III.1/+*). High magnification views of cells at the indicated stages are shown. In S5 spermatocytes (S5), UNO-EGFP signals are maximal in subnucleolar foci. After adjusting display settings to reveal low intensity EGFP signals (lower panel), fine dot signals are also apparent on autosomal chromatin (arrows). Similarly, in prometaphase I (prometa I) a single bright UNO-EGFP dot marks the sex chromosome bivalent, while far weaker dots are apparent on autosomal bivalents (lower panel, arrows). Scale bars = 25 μm (A) and 5 μm (B,C).

*EGFP* transgene expresses a functional UNO-EGFP protein. The EGFP signals detected in testis from *g-EGFP-uno* and *g-uno-EGFP* males were essentially identical to those obtained with anti-UNO with regard to subcellular localization and pattern of expression during spermatogenesis (Fig 3C). However, with *g-uno-EGFP* in a *uno* null mutant background, detection sensitivity appeared to be higher compared to immunolabeling with anti-UNO. Apart from the strong UNO-EGFP signals in the spermatocyte nucleoli and on sex chromosome bivalents during early M I, far weaker chromatin-associated dot signals were detected in the autosomal chromosome territories of spermatocytes as well as on autosomal bivalents during early M I (Fig 3C). These autosomal dots were just marginally above a more diffuse signal and detected in about 60% of the S5 spermatocytes. With anti-UNO, these autosomal signals were more difficult to distinguish clearly from non-specific background signals. In contrast to testis, we were unable to detect UNO-EGFP signals in ovaries.

In conclusion, UNO was found to behave as expected for an AHC protein. UNO is present on chromatin during the stages when bivalents are formed, and persists until the early stages of M I, as long as bivalents are maintained.

To determine pattern of expression and subcellular localization of the UNO-related protein encoded by *CG32117*, we generated transgenic lines expressing an EGFP fusion protein under control of the *CG32117* cis-regulatory region (*g-CG32117-EGFP*). According to transcript profiling data [51,52], *CG32117* mRNA is present at high levels exclusively in testis. Indeed, EGFP signals were readily detectable in *g-CG32117-EGFP* testes, but not until after the meiotic divisions (S6 Fig). In postmeiotic cysts with elongated spermatids, EGFP signals were observed in the elongated nuclei as well as throughout the elongated tails. While the apparent absence of pre-meiotic *CG32117* expression argued against a role in AHC, it appeared conceivable that the CG32117 protein product might inhibit adverse post-meiotic activities of AHC proteins in a dominant-negative manner. However, premature expression of CG32117-EGFP already in early spermatocytes, enforced with the driver *bam-GAL4-VP16* and a *UASt-CG32117-EGFP* transgene, did not result in a localization corresponding to that of the AHC proteins and did not appear to affect spermatogenesis. Overall, these observations argue against a functional involvement of *CG32117* in AHC.

## UNO co-localizes with SNM and MNM in an interdependent manner

The pattern of expression and the subcellular localization of UNO described above appeared to be identical to what has been previously reported in case of MNM and SNM [35]. For a careful comparison of the localization of UNO with that of MNM and SNM, we performed double labeling experiments. Anti-UNO antibodies were used for immunolabeling of testes (*bam>mnm-EGFP* and *bam>snm-EGFP*) expressing either MNM-EGFP or SNM-EGFP (Fig 4A and 4B). The anti-UNO signals were observed to co-localize with the EGFP signals. Co-localization was most extensive in late spermatocytes (S5) and at the start of the first meiotic division in case of the strong sex chromosome-associated signals in the subnucleolar foci and on the condensed bivalents, respectively (Fig 4A and 4B). Co-localization was also apparent on autosomal chromatin (Fig 4B), although these far weaker signals were not clearly above background in every spermatocyte.

To address whether UNO localization depends on *mnm* and *snm* gene function, we crossed the *g-uno-EGFP* transgene into *mnm* and *snm* mutant backgrounds (*mnm*$^{z3-3298}$/*mnm*$^{z3-5578}$ and *snm*$^{z3-2138}$/*snm*$^{z3-3426}$). In *snm* mutants, UNO-EGFP was not detected, suggesting that UNO protein is extremely unstable in the absence of SNM (Fig 4C). In contrast, in *mnm* mutants, where SNM protein is still present [35], UNO-EGFP was also clearly detectable in spermatocytes (Fig 4C). However, the subcellular localization of UNO-EGFP was not entirely normal. While still enriched in the nucleolus, UNO-EGFP was present in a rather diffuse manner without displaying the prominent subnucleolar foci characteristically observed in *mnm*$^+$ control animals. Moreover, during the initial stages of the first meiotic division, UNO-EGFP was not present in chromosome-associated dots (Fig 4C). Thus, MNM appears to be required for stable chromatin association of UNO.

While *tef* function is not required for localization of MNM and SNM on the sex chromosome bivalent and for normal sex chromosome conjunction, it is required for autosomal localization of MNM and SNM and for autosomal conjunction [33,35]. To evaluate the role of *tef* in UNO localization, we crossed the *g-uno-EGFP* transgene into *tef* mutants (*tef*$^{z2-3455}$/ *tef*$^{z2-4169}$). UNO-EGFP was readily detectable and displayed the normal strong nucleolar enrichment with prominent subnucleolar foci in *tef* mutant spermatocytes (Fig 4C). In contrast, the UNO-EGFP signal dots on autosomes that were detected in late control spermatocytes (Fig 3C) were absent in *tef* mutants (Fig 4C). As already emphasized above, the autosomal UNO-EGFP dots were marginally above background. They could not be detected unequivocally in all spermatocytes and all autosomal chromosome territories. However, a striking

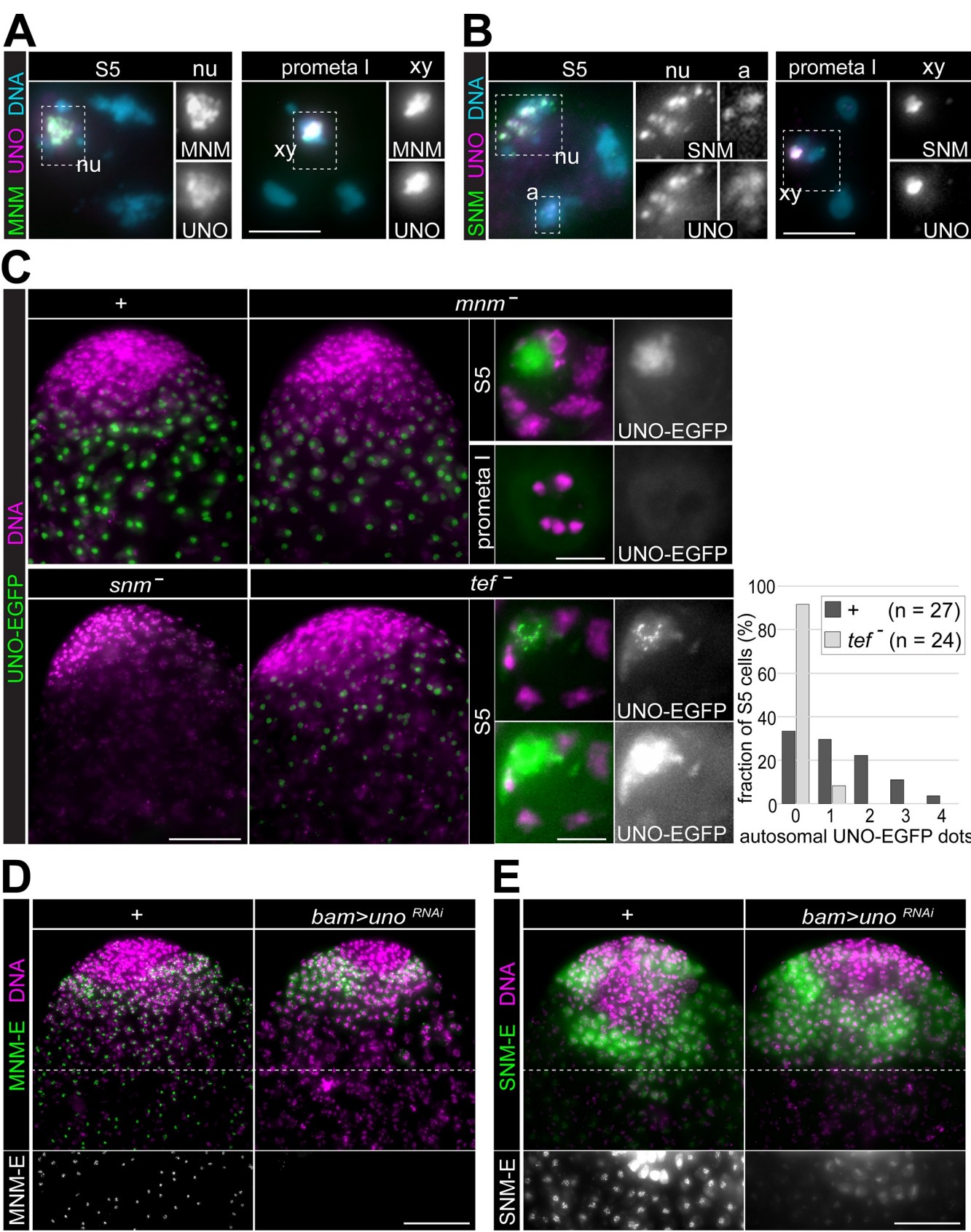

**Fig 4. UNO co-localizes with SNM and MNM in an interdependent manner. (A,B)** Squash preparations of testes from either **(A)** *bam>mnm-EGFP* or **(B)** *bam>snm-EGFP* males were labeled with anti-UNO and a DNA stain. High magnification views of cells at the indicated stages are displayed. As further illustrated with insets, co-localization of UNO with MNM-EGFP and SNM-EGFP is evident during S5 in subnucleolar foci (nu) and during prometaphase I in a dot associated with the sex chromosome bivalent (xy). In those S5 cells, which had detectable dot-like signals within autosomal territories (a), UNO and SNM-EGFP were also largely co-localized. **(C)** Localization of UNO-EGFP expressed from a transgene under control of the *uno* cis-regulatory region (*g-uno-EGFP*) was analyzed in control (+) as well as in *mnm (mnm⁻)*, *snm (snm⁻)* and *tef (tef⁻)* mutant testes. Testis tips are displayed in the four panels on the left side and high magnification views of single cells at the indicated stages on the right side. The number of UNO-EGFP dot signals in autosomal chromosome territories of S5 spermatocytes detected in control and *tef* mutants is presented in a histogram. **(D,E)** Localization of **(D)** MNM-EGFP and **(E)** SNM-EGFP in testes without (+) or with spermatocyte-specific UNO depletion (*uno*^RNAi). The *bamP-GAL4-VP16* driver in combination with *UASt* transgenes was used for expression of MNM-EGFP, SNM-EGFP and UNO depletion. Testis tips after a merge of the DNA and EGFP signals are displayed. The regions below the dashed white lines are also shown in the bottom row in order to document the lower EGFP signals in more mature spermatocytes, in (E) after display adjustment for further enhancement of weak signals. Scale bars = 5 μm (A, B, and C in right panels) and 50 μm (D, E and C left panels).

difference in the number of detectable UNO-EGFP dots on autosomal chromatin was observed after a careful comparison of identical squash preparations of *tef* mutant and control testes (Fig 4C). We conclude that *tef* is required for UNO localization on autosomal bivalents, while *tef* is dispensable for the strong UNO dot on the sex chromosome bivalent.

To evaluate whether *uno* function is required for the localization of MNM and SNM, we used *bam-GAL4-VP16* for spermatocyte-specific expression of *UASt-uno*^RNAi in combination with *UASt-mnm-EGFP* or *UASt-snm-EGFP*. While MNM-EGFP signals persisted throughout spermatocyte maturation in control testis without the *UASt-uno*^RNAi transgene (Fig 4D) [50], MNM-EGFP was detected only transiently in early spermatocytes when the *UASt-uno*^RNAi transgene was present (Fig 4D). Therefore, MNM protein appears to be unstable in the absence of UNO, reminiscent of the MNM absence in *snm* mutants [35]. Similar observations were made in case of SNM-EGFP, which is also detectable throughout spermatocyte maturation after *bam-GAL-VP16*-driven expression in control testis although with decreasing intensity in particular outside of subnucleolar foci (Fig 4E) [50]. In UNO-depleted spermatocytes, the initial SNM-EGFP expression was not affected, but thereafter signals decreased far more rapidly and were absent in late spermatocytes (Fig 4E).

In conclusion, consistent with the observed co-purification of UNO with MNM and SNM, we observed a precise co-localization of these proteins in particular in late spermatocytes and during M I. Moreover, the stable association of these AHC proteins with bivalent chromosomes depends on the presence of all three proteins. We suggest that UNO functions together with MNM and SNM in an AHC protein complex. The stable chromosomal association of this complex with autosomal bivalents requires TEF, which appears to be expressed at low levels that have escaped microscopic detection so far [34]. However, based on its co-purification with MNM-EGFP, TEF might be present in a subset of the AHC complexes.

## Separase cleaves UNO at the onset of anaphase I

In eukaryotes, the endoprotease separase allows sister chromatid separation during mitotic divisions by proteolytic cleavage of Scc1/Rad21, the α kleisin subunit of cohesin [9–11]. During canonical meiosis, separase cleaves a meiosis-specific α kleisin, Rec8, in a regionally controlled manner, on chromosome arms during M I and around centromeres during M II [27–31]. During the non-canonical meiosis in *Drosophila* males, the AHC proteins MNM and SNM are not co-localized with SMC1, a cohesin core subunit [35]. Nevertheless, AHC release at anaphase I onset requires the activity of *Drosophila* Separase (SSE) [48]. After spermatocyte-specific depletion of THR, an essential activator of SSE [57,58], bivalents are no longer separated into univalents at the onset of anaphase I, resulting in massive chromosome bridging during exit from M I [48]. Moreover, MNM and SNM protein, which normally lose their chromosomal association rapidly during the metaphase to anaphase transition of wild-type M I

[35,48], retain their chromosomal association during exit from M I after depletion of THR [48]. To assess the dynamics of UNO's chromosomal localization during M I and its dependence on separase activity, we performed time-lapse imaging with spermatocytes expressing His2Av-mRFP and UNO-EGFP instead of endogenous UNO. Without concomitant THR depletion, chromosomal UNO-EGFP signals were found to disappear very rapidly during exit from M I (Fig 5A; S2 Movie). In case of the strong UNO-EGFP dot marking the pairing site of the sex chromosomes, the disappearance could be readily quantified. Dot intensity started to drop at around the time when the autosomal bivalents appeared to have separated completely and therefore started to move rapidly and steadily towards the spindle poles (Fig 5C, S7 Fig). The onset of poleward chromatin movements was actually observed already 1–2 minutes earlier, but these initial movements were considerably slower, resulting primarily in a more stretched appearance of the bivalents, presumably reflecting the presence of residual linkage between homologs. During the rapid poleward segregation of the autosomes, the XY-associated UNO-EGFP dot intensity decreased briskly (Fig 5A; S2 Movie). Eventually, the fading XY dot was split into two remnant dots that vanished while moving along with the separating sex chromosomes to the spindle poles (Fig 5A; S2 Movie). On average, UNO-EGFP intensity within the XY-associated dot was reduced to background levels within about nine minutes (Fig 5C). At display settings revealing the weakest specific EGFP signals (Fig 5A, bottom row; S3 Movie), autosomal UNO-EGFP dots were clearly detectable during the initial M I stages in cells close to the coverslip. These autosomal UNO-EGFP signals were rapidly reduced to background levels around the time when the autosomes started to move rapidly and steadily towards the spindle poles. Therefore, this loss of weak autosomal UNO-EGFP dots occurred at the time when linkage between autosomes was lost.

After expression on *UASt-uno-EGFP* in a *uno*⁺ background, we observed the same UNO-EGFP behavior during M I by analogous time lapse analyses (S7 Fig, S4 Movie) as in the *uno* null background (Fig 5A). Moreover, additional analyses indicated that the meiotic UNO-EGFP behavior was largely identical to that of MNM-EGFP. For time-lapse analyses of MNM-EGFP, we used a transgene under *hsp83* promoter control [35], studying its basal expression. The resulting, relatively low levels of MNM-EGFP were readily compatible with simultaneous imaging of Cenp-A/CID-EGFP on centromeres. Thereby, the dynamics of the disappearance of the MNM-EGFP dot from the sex chromosome bivalent could be compared very accurately with the dynamics of bivalent separation (S7 Fig, S5 Movie). As with UNO-EGFP, the disappearance of MNM-EGFP from the sex chromosome bivalent started concomitantly with the rapid steady poleward movements of autosomes, with sex chromosome separation lagging slightly behind (S7 Fig, S5 Movie). We conclude that UNO disappears from bivalents during the metaphase to anaphase I transition just like the previously characterized AHC proteins MNM and SNM.

To clarify whether the disappearance of UNO from bivalents during exit from M I is dependent on separase activity, as previously shown for MNM and SNM [48], we completed analogous time-lapse imaging of UNO-EGFP after spermatocyte-specific THR depletion. Indeed, the disappearance of chromosomal UNO-EGFP signals was found to depend on separase activity (Fig 5B and 5C; S6 Movie). Quantification of UNO-EGFP signals in the XY-associated dot indicated that THR depletion resulted in a complete inhibition of the normal rapid disappearance during M I (Fig 5C). In addition, THR depletion resulted in a strong inhibition of bivalent separation during M I (Fig 5B, S6 Movie), as reported previously [48]. The massive chromosome bridge that was formed in parallel with chromosome decondensation during exit from M I was often cut by cytokinesis eventually, and a single strong UNO-EGFP dot signal remained clearly detectable throughout interkinesis (Fig 5B).

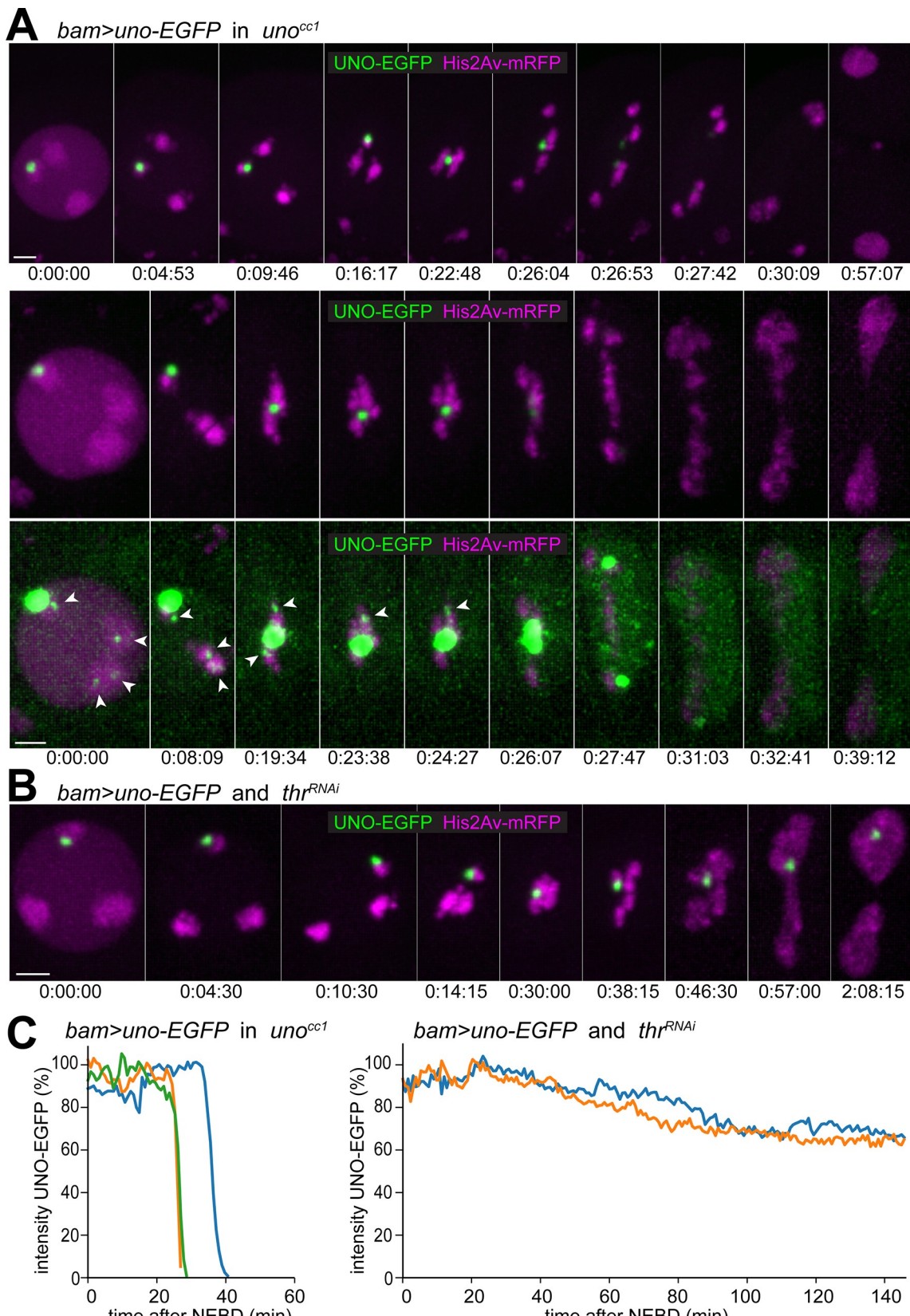

**Fig 5. UNO release from bivalents at the metaphase to anaphase I transition is separase-dependent. (A)** Progression through M I was analyzed by time-lapse imaging of *His2Av-mRFP* spermatocytes expressing UNO-EGFP instead of UNO (*bam>uno-EGFP* in *uno*[cc1] homozygotes). Still frames displayed in the top row represent the following stages (from left to right): just before onset NEBD, during NEBD, end prometaphase A, prometaphase B, metaphase, three consecutive time points during anaphase, telophase, and early interkinesis. Still frames in the lower rows are from a spermatocyte with well-detectable autosomal UNO-EGFP signals. In the bottom row, display settings were adjusted to reveal these weak autosomal signals (arrowheads). The following stages are shown (from left to right): just before onset NEBD, early prometaphase B, late prometaphase B, last metaphase frame, first anaphase frame, followed by frames during exit from M I. Time (hours:minutes:seconds) after the last time point before onset of NEBD I is indicated. **(B)** Progression through M I was analyzed by time-lapse imaging of *His2Av-mRFP* spermatocytes expressing UNO-EGFP and a *thr*[RNAi] transgene to eliminate separase activity (*bam>uno-EGFP* and *thr*[RNAi]). The still frames represent the following stages (from left to right): just before onset NEBD, during NEBD, end prometaphase A, prometaphase B, last metaphase frame, late anaphase, telophase, early and end of interkinesis. Time indicated as in (A). **(C)** Quantification of UNO-EGFP signal intensity in the XY-associated dot during progression through M I in spermatocytes of the indicated genotypes. Scale bars = 3 μm.

Previous findings regarding MNM and SNM [48] and those concerning UNO presented above suggested that these AHC proteins need to be eliminated in a separase-dependent manner to permit bivalent separation during M I in spermatocytes. Since the chromosomal localization of these colocalized AHC proteins is distinct from that of the core cohesin subunit SMC1 [35], it appeared unlikely that separase triggers bivalent separation by cleaving an α kleisin protein. Which alternative protein might be the relevant separase target during *Drosophila* male meiosis I? Plausible candidate targets are of course the AHC proteins. Therefore, we searched for putative separase cleavage sites that were conserved in the drosophilid AHC orthologs. Interestingly, UNO was found to have a highly conserved match to the separase cleavage site consensus ([E/D]XXR; X = any residue) [11,16,17,59] within the otherwise very divergent central linker domain (Fig 6B, S2 Fig).

To address whether the putative SSE cleavage site in UNO is functional, we generated *Drosophila* S2R+ cell lines expressing UNO variants which were inspired by the design of a previously described biosensor for separase activity in HeLa cells [60] (Fig 6A). In this original biosensor, a red and a green fluorescent protein domain were connected by the central region of the α kleisin Rad21, which includes separase cleavage sites. To establish biosensor functionality in *Drosophila* cells, we used the corresponding *Drosophila* Rad21/Vtd linker region and generated an S2R+ cell line expressing a similar fusion protein that was targeted to the centromere because it included the Cenp-A/Cid protein sequence N-terminally (Fig 6A). Time lapse imaging revealed that centromeres, which had both red and green signals before anaphase onset, rapidly lost the green signals during anaphase (Fig 6C), as expected [60]. Importantly, this specific loss of exclusively the green centromeric signals during anaphase was not observed in control cells expressing a biosensor protein with mutant separase cleavage sites (Fig 6B and 6C). These results confirmed that separase-mediated cleavage is detectable by our imaging assay with S2R+ cells. Therefore, we generated S2R+ cell lines expressing a biosensor variant in which the Rad21 linker region was exchanged by the UNO protein sequence. The wild-type UNO sequence was present in a first cell line, and in a second cell line a variant UNO sequence (UNO[nc]) with inactivating missense mutations in the separase cleavage consensus motif (Fig 6B). In case of the wild-type UNO sequence, we clearly observed a specific loss of the green centromeric signals during anaphase (Fig 6D). In contrast, the mutations in the putative separase cleavage site of UNO prevented the loss of green centromeric signals during anaphase (Fig 6D). These results indicate that SSE can cleave UNO, depending on the functionality of the identified separase cleavage motif.

## Uno cleavage is required for homolog separation

To evaluate the physiological significance of UNO cleavage by SSE, we compared the phenotypic consequences of *bamP-GAL4* driven expression of either *UASt-uno-EGFP* or *UASt-uno*[nc]*-EGFP* in a *uno* null mutant background. We selected *UASt* transgene insertions that resulted in

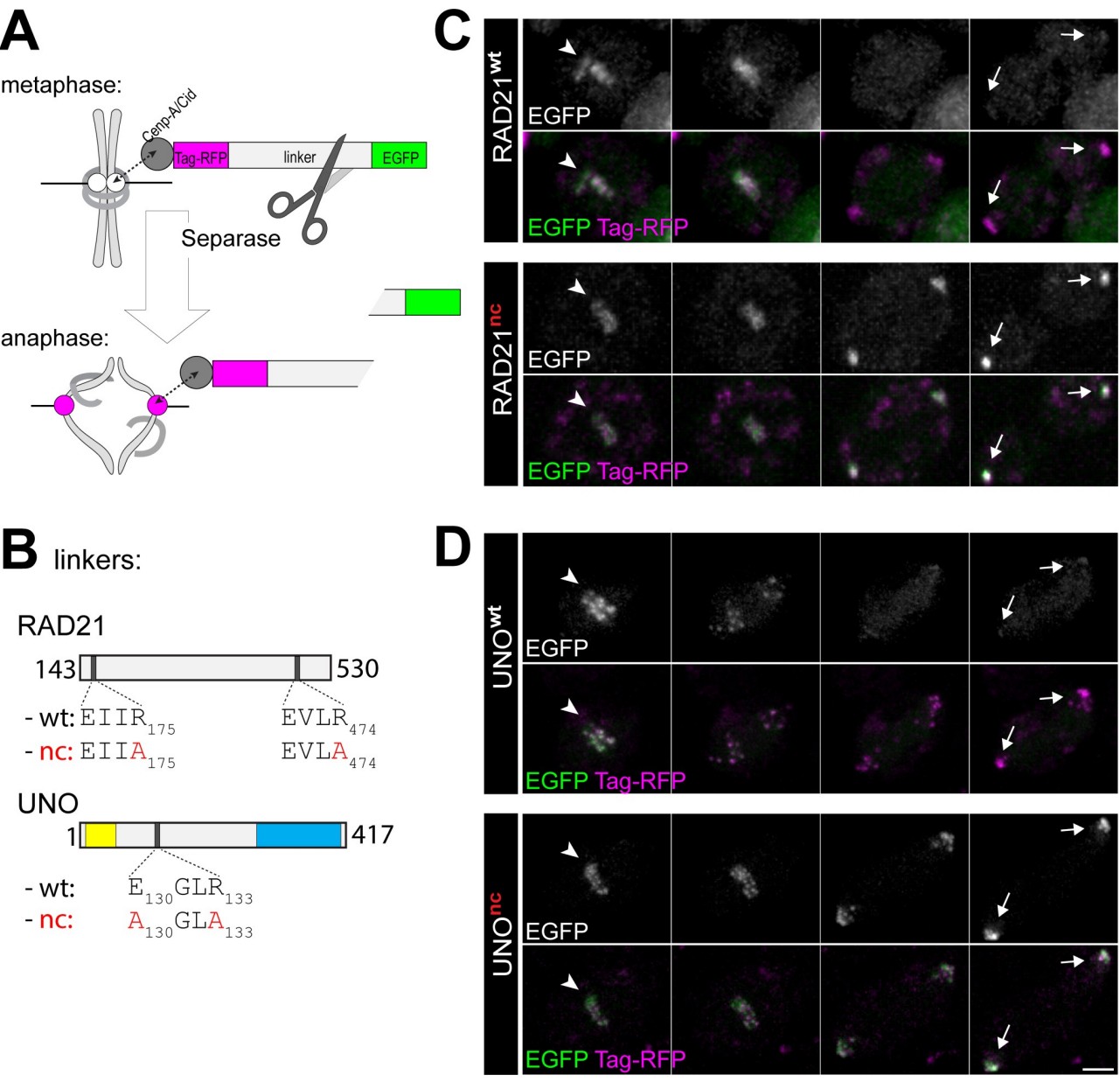

**Fig 6. UNO is cleaved by separase during the metaphase to anaphase transition. (A)** Scheme illustrating the separase cleavage assay with S2R+ cell lines. The assay involves expression of reporter proteins comprised of Cenp-A/Cid for centromere targeting followed by two distinct fluorescent proteins (Tag-RFP and EGFP) that are linked by a region with bona fide or putative separase cleavage sites. If functional separase cleavage sites are present, EGFP is cleaved off during the metaphase to anaphase transition because of endogenous separase activation. Accordingly, a loss of centromeric EGFP but not Tag-RFP signals during anaphase is detected by time lapse imaging. **(B)** Scheme illustrating the linker regions present in the analyzed reporter proteins. The linkers were from RAD21 (aa 143–530) containing either the known functional separase cleavage sites (wt) or the non-cleavable mutant versions (nc). In addition, full length UNO without (wt) and with mutations in the putative separase cleavage site (nc) were used as linkers. **(C, D)** Time lapse imaging was applied for analysis of centromeric Tag-RFP and EGFP fluorescence in cells expressing **(C)** either the RAD21$^{wt}$ or the RAD21$^{nc}$ reporter proteins, and **(D)** either the UNO$^{wt}$ or the UNO$^{nc}$ reporter proteins. The displayed still frames correspond to the last metaphase time point and the three consecutive time points acquired at 4 min intervals (from left to right). Centromere positions are indicated by arrowheads in metaphase (first still frame) and by arrows in late anaphase (last still frame). Scale bar = 5 μm.

equal expression levels of UNO-EGFP and UNO$^{nc}$-EGFP, respectively, as indicated by microscopic EGFP signal quantification. In addition, a *His2Av-mRFP* transgene was present for time-lapse imaging. As already described above, normal meiotic chromosome segregation was

restored by the *UASt-uno-EGFP* transgene (Fig 2C, Fig 5A). Similarly, until onset of anaphase I, spermatocytes expressing *UASt-uno^nc^-EGFP* instead of the endogenous *uno* were normal (Fig 7A and 7C). Just like UNO-EGFP, UNO$^{nc}$-EGFP was strongly enriched in a dot on the XY bivalent and weakly on autosomes (Fig 7A and 7C). Moreover, UNO$^{nc}$-EGFP also restored homolog conjunction. Normal bivalents were detected during prometaphase I after UNO$^{nc}$-EGFP expression in *uno* null mutants (Fig 7A and 7C), rather than univalents as in *uno* null mutants (Fig 2B and 2F, S1 Movie). Importantly, however, chromosome separation during anaphase I did not succeed in the presence of UNO$^{nc}$-EGFP. Instead, massive chromosome bridging was observed during anaphase I (Fig 7A and 7C), often followed by cutting of the decondensing chromatin by cytokinesis during telophase I (Fig 7C). Moreover, the chromosomal UNO$^{nc}$-EGFP signals did not disappear during exit from M I, remaining detectable during interkinesis (Fig 7A, 7B and 7C), M II and in early spermatids.

Time (hours:minutes:seconds) indicated in (A, C, and D) are relative to the last time point before onset of NEBD I and in (E) relative to the last time point before onset of metaphase I. Scale bars = 3 µm.

The inhibitory effect of UNO$^{nc}$-EGFP on bivalent separation during M I is predicted to be dominant. Therefore, we also analyzed progression through M I with *His2Av-mRFP* spermatocytes expressing UNO$^{nc}$-EGFP in a *uno$^+$* background (*bam>uno$^{nc}$-EGFP*). Bivalent separation was also affected in these spermatocytes (Fig 7D and 7E, S9 Movie and S10 Movie). But the inhibition of bivalent separation was not as complete as after UNO$^{nc}$-EGFP expression in *uno* mutants. Analysis of 20 spermatocytes from five distinct movies revealed that the inhibition of bivalent separation had a pronounced chromosome specificity. The large autosomal bivalents were usually observed to separate still normally after only a slight delay, while the chr XY bivalent consistently failed to separate during exit from M I, lagging in the equatorial region, and often forming a separate nucleus in addition to two daughter nuclei containing the separated autosomes (Fig 7D, S9 Movie). The strong UNO$^{nc}$-EGFP dot associated with the sex chromosome bivalent did not disappear during exit from MI (Fig 7D, S9 Movie). Similarly, the weak autosomal UNO$^{nc}$-EGFP dots did not disappear completely during exit from M I (Fig 7E, S9 Movie and S10 Movie) in contrast to the autosomal UNO-EGFP dots (Fig 5A). However, the separation of large autosomal bivalents during anaphase was accompanied by a transient stretching of the UNO$^{nc}$-EGFP dot, as well as by a weakening of its intensity (Fig 7E, S10 Movie).

In conclusion, our results indicate that separase cleaves UNO during the metaphase to anaphase transition of normal M I to release the conjunction between homologs, permitting their poleward separation during anaphase.

## Discussion

Until onset of anaphase I, homolog pairing needs to be maintained. During canonical meiosis, maintenance of homolog pairing is achieved by cross-overs and distal sister chromatid cohesion. However, an absence of cross-overs during meiosis in *Drosophila* males was reported long ago [61]. Therefore, homolog conjunction until exit from M I must rely on an alternative mechanism in *Drosophila* spermatocytes. Genetic approaches have identified the three proteins MNM, SNM and TEF, which are specifically required for this alternative homolog conjunction (AHC) [33–35]. The biochemical mechanisms whereby these proteins contribute to AHC are not understood. Moreover, while separase was shown to be required for AHC resolution at anaphase I onset [48], the critical target was not identified. For further clarification of how AHC is established and resolved during *Drosophila* male meiosis, we have chosen a biochemical approach. By analysis of proteins co-purified with MNM-EGFP and SNM-EGFP, we

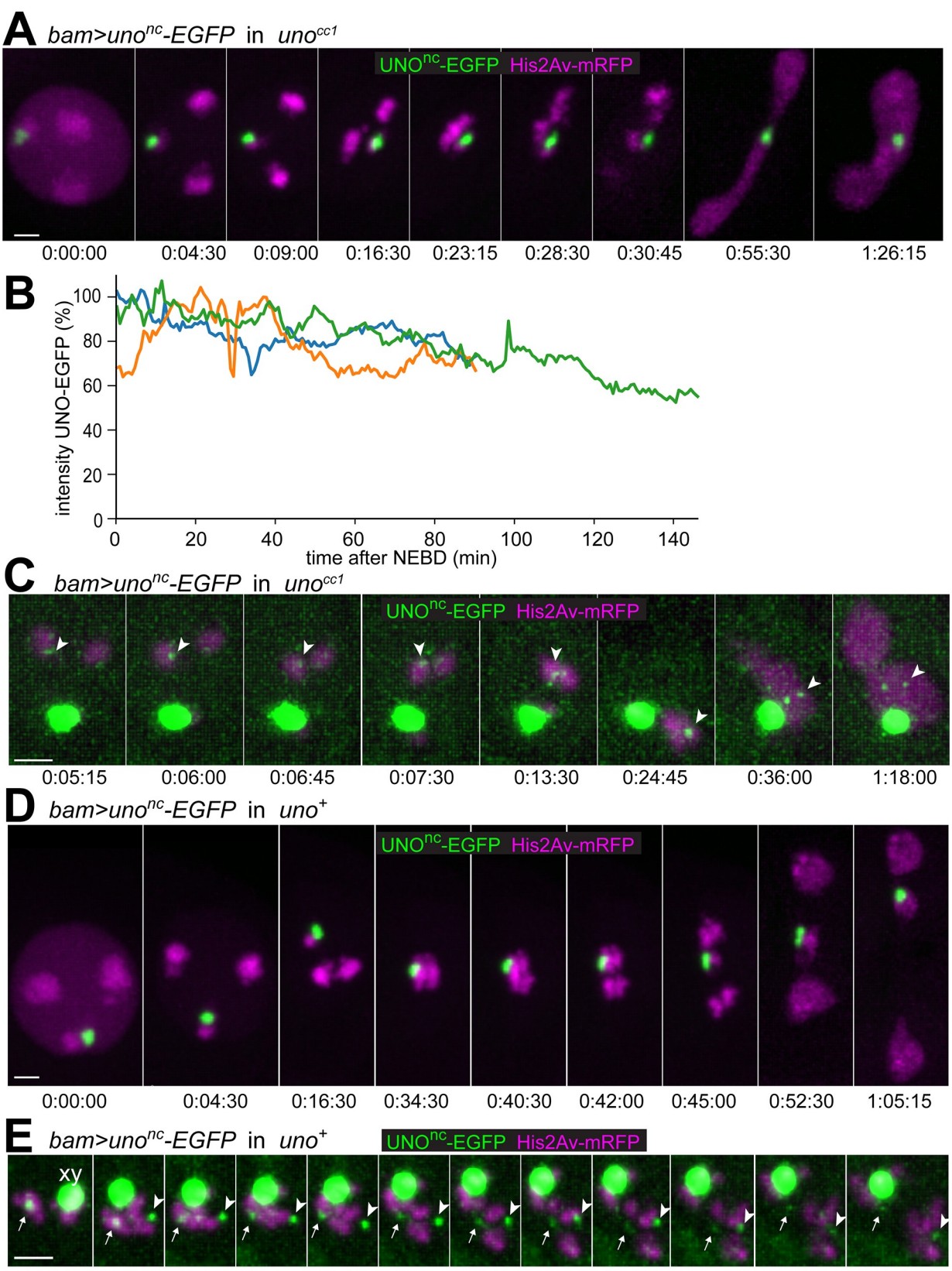

**Fig 7. Separation of bivalents during M I is blocked by mutational inactivation of the separase cleavage site in Uno. (A)** Progression through M I was analyzed by time-lapse imaging of *His2Av-mRFP* spermatocytes expressing UNO$^{nc}$-EGFP instead of UNO (*bam>uno$^{nc}$-EGFP* in *uno$^{cc1}$*). Stages shown (from left to right): just before onset NEBD, during NEBD, end prometaphase A, prometaphase B, metaphase, early and late anaphase, telophase, and interkinesis. **(B)** Signal intensity of UNO$^{nc}$-EGFP in the dot associated with the sex chromosome bivalent was quantified in three spermatocytes expressing UNO$^{nc}$-EGFP instead of UNO (*bam>uno$^{nc}$-EGFP* in *uno$^{cc1}$*), as shown in (A). The intensity curves, aligned at the time point just before onset of NEBD (t = 0), indicate that the rapid disappearance that occurs in case of UNO-EGFP between t = 30–40 minutes does not occur in case of UNO$^{nc}$-EGFP. **(C)** Display settings were adjusted to reveal the weak autosomal signals (arrowheads) in a spermatocyte expressing UNO$^{nc}$-EGFP instead of UNO (*bam>uno$^{nc}$-EGFP* in *uno$^{cc1}$*). Stages shown (from left to right): four consecutive time points from prometaphase A, metaphase, telophase, and interkinesis. **(D,E)** Progression through M I was analyzed by time-lapse imaging of *His2Av-mRFP* spermatocytes expressing UNO$^{nc}$-EGFP in an *uno$^+$* background (*bam>uno$^{nc}$-EGFP* in *uno$^+$*). **(D)** Stages shown (from left to right): just before onset NEBD, during NEBD, prometaphase B, metaphase just before onset of bivalent stretching, just before final separation of autosomal bivalents, early during poleward transport of separated autosomes, later during poleward transport of autosomes, telophase, and interkinesis. **(E)** The region containing the bivalents of a spermatocyte, in which autosomal UNO$^{nc}$-EGFP dots were well detectable, during the metaphase I to anaphase I transition. Display settings were adjusted to reveal the weak dot signal associated with one of the large autosomal bivalents (arrows) and with the chr 4 bivalent (arrowheads). The strongly saturated dot signal associated with the sex chromosome bivalent (xy) is labeled in the first still frame.

have identified UNO, a novel protein. Our characterization demonstrates that UNO is required for AHC. UNO is co-localized with MNM and SNM on bivalents in an inter-dependent manner. Moreover, UNO is cleaved by separase, and mutations that prevent this cleavage abolish bivalent separation during exit from M I. Therefore, we propose that separase-mediated cleavage of this novel AHC protein UNO triggers bivalent separation at anaphase I onset.

Our biochemical approach relies on a newly developed protocol for mass isolation of testes from pupae and late third instar wandering stage larvae. During these stages, the testes are lightly embedded within the fat body. In addition, they are not yet attached to the distal part of the reproductive tract, which develops only after pupariation from the genital imaginal disc. Thus, the late larval/early pupal testes can be dissociated gently in intact form away from other organs and tissues. The number of spermatocytes per testis is around 800–1000 during pupariation. Spermatocyte number is not significantly higher in adult testes, which however also contain late spermatid stages that are not yet present around pupariation. The great majority of published reports describing biochemical approaches with *Drosophila* testis (like RNA-Seq, ChIP-Seq, shotgun proteomics or AP-MS/MS) were accomplished after manual dissection of up to a few thousand testes. Our mass isolation protocol readily provides at least tenfold more material. It should greatly facilitates future biochemical as well as omics-based analyses of *Drosophila* spermatogenesis.

Beyond UNO, a considerable number of proteins were co-purified efficiently with MNM-EGFP and SNM-EGFP, but not or at most marginally in the EGFP control. While it is not excluded that some of these proteins act as physiological interaction partners in AHC, the known properties of some of these putative interactors are difficult to reconcile with such a role. Several (like Ankyrin 2, alpha-Spectrin, beta-Spectrin, Zipper, Spaghetti squash, Tropomodulin and Futsch) are known to function as cytoskeletal elements outside of the nucleus. In this context, two potential problems with our EGFP control purifications might be relevant. Because of distinct regulatory sequences (from the *bam* and *exu* genes, respectively), the expression pattern of MNM-EGFP and SNM-EGFP on the one hand and EGFP on the other hand were not identical. The expression of the EGFP-tagged AHC proteins is likely maximal already in early spermatocytes, which are only starting with EGFP control protein accumulation. Moreover, the EGFP-tagged AHC proteins are also expressed in ovaries, in contrast to the EGFP control. We cannot exclude a presence of contaminating ovaries in case of the MNM-EGFP and SNM-EGFP purifications. However, EGFP signal intensities resulting from bam-GAL4-VP16 driven expression of MNM-EGFP or SNM-EGFP in ovaries are very low and most likely insufficient for retention during the BioSorter purification step.

UNO like SNM, TEF and perhaps MNM appear to be encoded exclusively in genomes of higher dipterans that use AHC during male meiosis instead of the canonical cross-over based

linkage of homologs. Ancestral genes, from which *mnm*, *snm* and *tef* might have evolved, have been identified. In contrast, the evolutionary origin of *uno* is not evident. The only gene with obvious sequence similarity to *uno* is *CG32117*. However, the distribution of *CG32117* orthologs is even more restricted than that of *uno*, suggesting that *CG32117* is derived from *uno* rather than being its ancestor. While *CG32117* expression is strictly testis-specific, our initial functional characterization suggests that is does not have a role in AHC. The CG32117 protein was not detected among the proteins co-purified with MNM-EGFP and SNM-EGFP. Its expression, as reported by the *g-CG32117-EGFP* transgene, started only well after meiosis, and mutations did not appear to have an effect on meiosis.

The fact that UNO includes a functional separase cleavage site argues for a potential relatedness with α kleisins. Many proteins contain matches to the separase cleavage site consensus, but only very few of these are true substrates [59]. Matches to the minimal consensus of separase cleavage sites ([E/D]XXR) can also be found in MNM and SNM, but reporter proteins containing these protein sequences as linkers failed to reveal cleavage during exit from mitosis in contrast to the analogous reporters with the UNO or RAD21 linkers. Beyond α kleisins [9–11], separase itself [62] and *Drosophila* THR [58], the regulatory separase subunit that corresponds to the N-terminal separase region in other species [57], are cleaved by separase. Even in α kleisins and separase, several additional matches to the cleavage consensus are not cleaved by separase. An independent separase docking site was recently shown to contribute to cleavage efficiency in the budding yeast α kleisin Scc1 [63]. Matches to the corresponding motif ([L/V/I/M]PE) are absent from UNO, THR, and MNM, but one is present in SNM.

In addition to α kleisins and separases, only three additional separase substrates have been identified (Slk19, kendrin/pericentrin, and CPAR-1) [59,64–66]. These additional substrates are not cohesin subunits. While these precedents demonstrate an existence of physiological separase substrates beyond the functional context of cohesion control, a relatedness between UNO and the α kleisins is supported by an additional fact. SNM, the AHC protein that is physically associated with UNO, is a member of the Scc3/SA/Stromalin protein family [35]. These proteins function as cohesin subunits and bind directly to α kleisins [1,46,47]. The Scc3/SA protein binding site in the Scc1/Rad21 α kleisins is located within the central, poorly conserved and largely unstructured region [1,47] that links the more strongly conserved and structured N- and C-terminal domains [67]. The central Scc1/Rad21 linker region also includes the separase cleavage sites [1,9–11]. The N-terminal domain of Scc1/Rad21 contains two conserved α-helices, which mediate binding to the Smc3 cohesin subunit by formation of a four helix bundle [1,8]. The C-terminal domain of Scc1/Rad21 forms a winged helix domain that binds to the head domain of Smc1 cohesin subunit [1,7]. These conserved and structured N- and C-terminal α kleisin domains cannot be recognized in UNO, suggesting that it does not bind to Smc proteins. Previous findings provide additional arguments against the notion that UNO and SNM function like the α kleisins and Scc3/SA proteins in cohesin complexes. The subcellular localization of SMC1 in *Drosophila* spermatocytes is clearly distinct from that of SNM and MNM [35], which in turn are precisely co-localized with UNO, as shown here. Moreover, genetic analyses have indicated that AHC does not depend on the meiotic cohesin subunit SOLO and SUNN [68,69], which were proposed to represent meiosis-specific *Drosophila* α kleisin and SA proteins, respectively [70,71]. While *snm* and perhaps *uno* have evolved from genes encoding cohesin complex subunits, their protein products do not appear to function as cohesin complex subunits.

While our identification and characterization of UNO clearly advances our mechanistic understanding of how the AHC is resolved on time for bivalent separation at the metaphase-to-anaphase I transition in *Drosophila* spermatocytes, the molecular explanation for how the AHC mediates linkage between homologous chromosomes beforehand is still rudimentary.

Our affinity-purification of MNM-EGFP and SNM-EGFP demonstrate that the AHC proteins are in physical association. Presumably, a protein complex containing MNM, SNM and UNO binds somehow to the known pairing site on the sex chromosomes, i.e., to the 240 bp repeats within the IGS of the rDNA repeats [38]. In case of autosomal bivalents, the TEF protein might join the protein complex with MNM, SNM and UNO, perhaps only transiently. In contrast, to MNM, SNM and UNO, which are detected in weak dots on autosomal bivalents during early M I after *bam-GAL4-VP16* driven expression of EGFP-tagged versions, TEF-EGFP is not detectable after analogous expression [34]. However, TEF was clearly co-purified with MNM-EGFP, indicating that at least these two AHC proteins are engaged in physical interactions. The weak autosomal dots formed by EGFP-tagged MNM, SNM and UNO suggest that AHC occurs also at restricted autosomal sites. However, genetic analyses have provided strong evidence against fixed unique pairing sites on autosomes, analogous to the chr XY pairing sites [39,40]. Therefore, autosomes were hypothesized to have many potential pairing sites, among which only one or a few might be selected and used quite randomly in an individual spermatocyte [40,72]. TEF, which appears to be present at lower levels than the other AHC proteins and perhaps only transiently, might be involved in this hypothetical selection of autosomal pairing sites, a process that would be analogous to cross-over designation among an excess of DSBs during canonical meiosis [20].

The autosomal AHC dots appear to contain far lower amounts of MNM, SNM and UNO than those associated with the chr XY pairing site. The autosomal AHC dots might also confer weaker homolog conjunction, as supported by the chromosome-biased effects of UNO[nc]-EGFP expression in an *uno[+]* background. While chr XY bivalent separation was completely inhibited in this case, large autosomal bivalents were usually separated eventually. Elimination of conjunction between large autosomal bivalents in the presence of UNO[nc]-EGFP might perhaps involve additional target sites that are cleaved by separase although with low efficiency only. Beyond the cleavage site that we have identified, UNO includes one additional match to the consensus of separase cleavage sites ([E/D]XXR), which is however far less conserved within drosophilids. Alternatively, conjunction might depend on oligomeric assemblies containing multiple UNO copies, and cleavage of one UNO subunit might already be sufficient for functional inactivation. In addition, the pulling forces exerted by the spindle during anaphase might contribute to some extent to a dissociation of chromosomal AHC protein complexes independent of cleavage by separase. The future clarification of these and many other unknowns in the molecular basis of AHC during *Drosophila* male meiosis will likely further profit from the testis mass isolation procedure described here.

## Material and methods

### *Drosophila* lines

The following lines with mutations or transgenes have been described before: *Df(2R)Exel7094* [73], *Df(3L)ED4486* [74], *mnm[Z3-3298]*, *mnm[Z3-5578]*, *snm[Z3-0317]*, *snm[Z3-2138]* [35], *tef[Z2-4169]* and *tef[Z2-3455]* [33], *hsp83-mnm-EGFP* [35], *His2Av-mRFP* and *gCid-EGFP-Cid* [75], *bamP-GAL4-VP16* [76], *UASt-snm-EGFP* and *UASt-mnm-EGFP* [50], *UAS-V20thr[shmiR9]* [48] and *gitdTomato2x-Cenp-C* [77]. For transgenic UNO-RNAi, we used *w[1118]*; *P{GD14007}* (v46108) [78]. Lines with the following transgenes were generated with the plasmid constructs described further below: *exumP-EGFP* (inserted into the landing site P{CaryP}attP40 on chromosome 2L), *g-EGFP-uno*, *g-uno-EGFP*, *UASt-uno-EGFP*, *UASt-uno[nc]-EGFP*, *g-CG32117-EGFP*, *UASt-CG32117-EGFP*. The *uno* null alleles (*uno[cc1]*, *uno[cc2]*, *uno[cc3]* and *uno[cc4]*) as well as the *CG32117* null alleles (*CG32117[cc1]*, *CG32117[cc2]*, *CG32117[cc3]* and *CG32117[cc4]*) were generated using CRISPR/cas essentially as described [79]. Briefly, embryos from *y[1] M{w[+mC] = Act5C-Cas9*.

*P}*ZH-2A *w\** (Bloomington *Drosophila* stock Center #54590) were injected (BestGene Inc., Chino Hills, CA, USA) with the pCFD5 derived plasmids described further below. Each of these pCFD5 derivatives allowed expression of a gRNA pair targeting two distinct gene regions. The established mutant *uno* and *CG32117* alleles were characterized by enzymatic amplification of a gene region with forward and reverse primers that annealed just upstream and downstream of the two gRNA targets sites, respectively. DNA sequencing of the amplified fragments confirmed the presence of intragenic deletions. Accordingly, the predicted UNO protein expressed from *uno*[cc1], the allele predominantly used for our phenotypic characterizations, is only 37 amino acids long with the two initial amino acids of UNO (MP) fused out of frame to 35 extra amino acids. The *uno* alleles cc2 –cc4 represent similar out-of-frame deletions that retain only a few N-terminal *uno* codons ahead of the deletion breakpoints. Similarly, the *CG32117* alleles cc1 –cc4 were found to have large intragenic out-of-frame deletions leaving around 20 codons of *CG32117* upstream of the deletion breakpoint. The fertility of *uno*[cc1]/ *Df(2R)Exel7094* was analyzed and compared to that of control flies (+/ *Df(2R) Exel7094*). For the analysis of male fertility, replicate crosses were set up for the two genotypes (10 and 8, respectively), each with a single male and three *w*[1] virgin control females. For the analysis of female fertility, replicate crosses were set up for the two genotypes (8 and 6, respectively), each with a pool of three virgins females and three *w*[1] control males. After pre-mating for a few days, crosses were turned over into fresh vials, parents were discarded after a few days and all eclosing adult F1 progeny was counted. Fertility of males lacking *CG32117* function was estimated using *w*[1118] as control. Three control virgins were crossed with three males carrying either the *CG32117* allele cc1, cc2, cc3, or cc4 over *Df(3L)ED4486* which deletes the *CG32117* gene. After pre-mating for a few days, crosses were turned over into fresh vials, parents were discarded after a few days and all eclosing adult F1 progeny was counted. Standard crossing was used for the generation of the various strains used for experimental analyses. The genotypes of the flies analyzed are described in detail in the supplemental material (S2 Table). All flies analyzed were raised at 25˚C.

## Plasmids

For the generation of the pCFD5 [79] derivatives (pCFD5-uno_1, pCFD5-uno_2, pCFD5-CG32117_1 and pCFD5-CG32117_2) we inserted synthetic double stranded (ds) DNA fragments (JW98-101) (Integrated DNA Technologies, Coralville, IA, USA) into the vector after digesting with BbsI. See supplemental material for the sequences of the dsDNA fragments (S3 Table).

The vector pCaSpeR5 was used for the generation of the transgene constructs pCaSpeR5-g-EGFP-uno and pCaSpeR5-g-uno-EGFP. Genomic fragments with *uno* upstream, coding and downstream regions were amplified from w1 genomic DNA using primer pairs JW88/JW105, JW106/JW91 and JW92/JW93, respectively. For all primer sequences, see supplementary material (S3 Table). The fragments were inserted sequentially into the vector using the restriction enzymes AgeI/ NgoMIV, NgoMIV/ SacII and SacII, respectively. The resulting intermediate (pCaSpeR5-g-uno) contains the *uno* genomic region with an endogenous NgoMIV site immediately downstream of the initiation codon and an introduced SacII site immediately in front of the stop codon. The EGFP coding region was amplified from UASt-mcs-EGFP [80] using primer the pair JW94/JW95 and inserted into the NgoMIV site, yielding pCaSpeR5-g-EGFP-uno. For pCaSpeR5-g-uno-EGFP, the EGFP coding sequence was amplified from UASt-mcs-EGFP [80] using the primer pair JW96/JW97 and inserted into the SacII site.

The transgene constructs pUASt-uno-EGFP and pUASt-uno[nc]-EGFP were generated with the vector pUASt-mcs-EGFP (). In case of pUASt-uno-EGFP, the *uno* coding region was

amplified from $w^1$ genomic DNA using the primers JW82/JW83 and inserted using the restriction sites BglII and XhoI. For pUASt-uno$^{nc}$-EGFP, we first introduced the E130A and R133A mutations into pCasper5-uno-EGFP by mutagenic plasmid amplification [81] using the primer JW118. The resulting *uno*$^{nc}$ coding region was amplified again using JW82/JW83 and the BglII and XhoI sites.

For the generation of pCaSpeR5-CG32117-EGFP, we amplified the *CG32117* region as two fragments from $w^1$ genomic DNA, the first fragment with 5' and coding region using the primer pair CL322/CL323 and the second fragment with the 3' region using primer pair CL324/CL325. The primers CL323 and CL 324 resulted in the introduction of an MluI site immediately upstream of the *CG32117* stop codon for insertion of the EGFP coding region. This coding region was amplified from pUASt-mcs-EGFP using CL326/CL327. The three fragments were inserted sequentially into pCaSpeR5 using the restriction sites NheI (fragment 1), MluI/ EcoRI (fragment 2) and MluI (EGFP). The final construct has 789 bp of genomic upstream sequence preceding the annotated transcriptional start site, followed by the CG32117-EGFP coding sequence and 276 bp of downstream sequences.

For the generation of pUASt-CG32117-EGFP, the *CG32117* coding region was amplified from $w^1$ genomic DNA using the primers CL320/CL321 and inserted into pUASt-mcs-EGFP using XhoI and EcoRI.

The *exumP-EGFP* transgene construct was generated with the vector pattB [82]. It contains upstream sequences from the *exuperantia* (*exu*) gene which is expressed in both the male and the female germline but from distinct promoters. We used a fragment (*exum*) with the promoter driving expression exclusively in the male germline. The *exum* fragments started and ended 1025 and 291 bp upstream of the *exu* initiation codon, respectively. After this regulatory region, the EGFP coding sequence was placed with an N-terminal extension of six amino acids. The first four of these amino acids (MVAD) correspond to the start of EXU followed by two amino acids (EF) introduced by an EcoRI site used for insertion of the EGFP coding region. The coding region was followed by 1216 bp of *exu* 3' sequences.

Plasmid constructs for the generation of stably transformed S2R+ cells expressing reporter proteins with linker regions containing functional or non-functional separase cleavage sites were made with the pMT-bla vector. This vector includes the $Cu2^+$-inducible *Metallothionein A* (*MtnA*) promoter and a blasticidin resistance cassette [83]. The complete *Cenp-A/cid* coding region (without stop codon) [75] was inserted downstream of the *MtnA* promoter. After the *cid* coding sequence, additional fragments were introduced in frame fusing the coding sequence of a happy linker [84] followed by those of Tag-RFP-T [85] and EGFP. The resulting intermediate construct pMT-cid-HaLi-TagRFPT-GFP had a NotI site between the sequences encoding the two fluorescent proteins. This NotI site was used for the introduction of linker regions derived from *Rad21/vtd* or *uno*. The wild-type *Rad21* linker region (with functional cleavage sites) was amplified from pCaSpeR{$w^+$, *αtub-Rad21-EGFP*} [86] and the mutant non-cleavable version from pUASP-Rad21$^{nc}$-myc [86]. For the *uno* reporter constructs, the full coding regions were amplified from either pUASt-uno-EGFP or pUASt-uno$^{nc}$-EGFP and inserted as linker regions into the NotI site of pMT-cid-HaLi-TagRFPT-GFP.

## Testis mass isolation

Flies of the appropriate genotypes (*bam>mnm-EGFP*, *bam>snm-EGFP*, *exum-EGFP*) were first expanded at 25°C by collection of eggs in bottles (4.7 cm diameter, 9.5 cm high with *Drosophila* food filling the first 2.5 cm). The next generation of adult flies from 25 bottles (about 5000 flies) were transferred into a large collection cage (~80 x 80 x 80 cm). Four plastic plates (diameter 14 cm, rim height 2 cm) containing fly food were placed in the cage for egg

collection. Every 24 hours, an additional round of egg collection with four fresh plates was started. After removing the plates with eggs from the cage, they were incubated for two days at 25°C. Thereafter, larvae and remaining fly food were transferred to plexiglass puparation cylinders (diameter 14 cm, height 50 cm). When necessary, additional food was added to the puparation cylinders. When the first brown pupae were forming on the wall inside the puparation cylinders, testis mass isolation was initiated. All pupae and wandering stage L3 larvae were removed from the cylinder wall (using tap water and a silicon scrapper) and collected in a sieve (700 µm pore size). The pupae/larvae were transferred to Ringer solution (182 mM KCl; 46 mM NaCl; 3 mM $CaCl_2$, 10 mM Tris-HCl, pH 7.2) and disrupted in a metal grain mill attachment for a standard food processor (KSM125 equipped with KGM, Kitchen Aid). A constant flow of Ringer solution was maintained during grinding and passed first through a sieve with 355 µm pore size. Material that did not pass this sieve was reapplied to the grinder. Material that passed this first sieve was passed through a second sieve with 100 µm pore size to retain testes and to reduce excess liquid. The retained material was rinsed from this second sieve into 50 ml Falcon tubes with Ringer solution. Based on analyses of samples removed at successive steps of the grinding procedure with a stereomicroscope equipped for fluorescence detection, we estimate that the gonads are released from close to 100% of the larvae/pupae and 50–80% of these released gonads appear to remain intact.

The next two steps, Ficoll density gradient centrifugation and fluorescence-based particle sorting, were performed as described [87] with the following adaptations. After a settling period of 15 minutes, the supernatant was discarded and the sediment was applied on top of a 2-layer Ficoll PM 400 gradient. The gradient was prepared by adding first 20 ml of 25% Ficoll in Ringer solution into a 50 ml Falcon tube, followed by 20 ml of 12% Ficoll in Ringer solution). The loaded gradient was centrifuged for 1 hour at 4°C at 4500 rcf. The part of the gradient between the two interphases (25%/12% and 12%/0%) including the material in the interphases was transferred to new tubes and diluted 1:5 in Ringer solution. After a settling period of 30 minutes, the supernatant was discarded. For fluorescence-based particle sorting, the remaining suspension was applied to a BioSorter (Union Biometrica; FOCA 1000; standard settings). Depending on particle density, the suspension was further diluted in Ringer solution for optimal sorting. The gates for sorting were set after measuring the following parameter of larval testes: time of flight, extinction, green fluorescence and red fluorescence (to estimate auto florescence). Sorted testes were recovered in a 50 ml tube with Ringer solution, transferred to a 1.5 ml Eppendorf tube and snap frozen in liquid nitrogen after settling of testes and removal of supernatant.

The purity of the final fraction depends critically on testis-specificity and level of the GFP signal, as well as on the precise parameter settings chosen for sorting with the BioSorter. In the optimal case, the fraction of contaminating tissue pieces other than testes can be as low as a few percent based on inspection with a stereomicroscope (S8 Fig). However, to an extent that varies among the testes, they still have residual amounts of fat body attached (S8 Fig). For mass isolation of the testes used in our AP-MS experiments, BioSorter parameters were optimized using larvae/pupae of the bam>MNM-EGFP genotype. In this genotype, GFP signals were lowest. The chosen parameters, representing a compromise between sorting speed and purity, were applied identically for testis mass isolation from the other genotypes. With these parameters, sorting of 5000 tissue particles lasted for 3 hours and about 80% of the sorted tissue particles were gonads. The residual 20% contaminating particles were of two types. About half of them displayed green fluorescence for unknown reasons. The other half did not emit green fluorescence. These latter particles were presumably co-purified along with a GFP-positive particle residing in the same drop. According to inspection by light microscopy, the contaminants were morphologically heterogeneous and did not appear to be derived from a defined

tissue. Based on a numerical comparison of larvae/pupae at the start of mass isolation and testes in the final sample, we estimate the overall testis yield to be about 15%.

## Affinity purification from testes extracts

For each of the four replicates per genotype, 6000–8000 testis were pooled and homogenized in 1.5 ml lysis buffer (20 mM Tris-HCl at pH 7.5; 150 mM NaCl; 2 mM MgCl$_2$; 0.1% Nonidet NP-40 [Sigma Aldrich IGEPAL CA-630]; 5% glycerol; 1 tablet /10 ml c0mplete protease inhibitor cocktail [Roche 04693159001]; 1 mM DTT; 50 U/ml benzonase) using a Potter-Elvehjem homogenizer with a PTFE pestle. Two clearing steps were performed, each involving a 15 minute centrifugation (16000 x g at 4˚C), followed by transfer of the supernatant to a new tube. 50 µl µMACS anti-GFP beads (Miltenyi Biotec, Solothurn, Switzerland, catalog number 130-091-125) were added and incubated for one hour on a rotating wheel. The solution was applied to a pre-wet µColumn (Miltenyi Biotec, catalog number 130-042-701) in the magnetic stand (Miltenyi Biotec, catalog number 130-042-602 and 130-042-303) and the column was washed four times with 200 µl lysis buffer and once with 200 µl Tris-HCl, pH 7.5. All steps were performed at 4˚ C in low binding tubes (Eppendorf, catalog number 0030108051) to minimize loss of proteins.

To prepare the sample for MS analyses we used the PreOmics iST kit (PreOmics GmbH, Planegg/Martinsried, Germany, article number P.O.00001) with the following transition steps. 25 µl of LYSE buffer (PreOmics iST kit) were added to the column. The column was removed from the magnetic stand and placed on top of a 1.5 ml tube. 50 µl of LYSE buffer were added to the column to elute the beads and proteins from the column onto the cartridge (PreOmics iST kit). Reduction and alkylation were performed at 60˚C for 10 minutes in a shaker at 1000 rpm. Digestion with Trypsin and Lys-C, as well as purification were performed according to the manufacturer's protocol.

## Mass spectrometry

Samples were injected by an Easy-nLC 1000 system (Thermo Scientific) and separated on an EasySpray-column (75 µm x 500 mm) packed with C18 material (PepMap, C18, 100 Å, 2 µm, Thermo Scientific). The column was equilibrated with 100% solvent A (0.1% formic acid (FA) in water). Peptides were eluted using the following gradient of solvent B (0.1% FA in acetonitrile): 5% B for 2 minutes; 5–25% B in 60 minutes; 25–35% B in 10 minutes; 35–95% B in 5 minutes at a flow rate of 0.3 µl/minutes. High accuracy mass spectra were acquired with an Orbitrap Fusion (Thermo Scientific) that was operated in data dependent acquisition mode. All precursor signals were recorded in the Orbitrap using quadrupole transmission in the mass range of 300–1500 m/z. Spectra were recorded with a resolution of 120'000 at 200 m/z, a target value of 5E5 and the maximum cycle time was set to 3 seconds. Data dependent MS/MS were recorded in the linear ion trap using quadrupole isolation with a window of 1.6 Da and HCD fragmentation with 30% fragmentation energy. The ion trap was operated in rapid scan mode with a target value of 8E3 and a maximum injection time of 80 ms. Precursor signals were selected for fragmentation with a charge state from +2 to +7 and a signal intensity of at least 5E3. A dynamic exclusion list was used for 25 seconds and maximum parallelizing ion injections was activated. After data collection the peak lists were generated using Proteome Discoverer 2.1 (Thermo Scientific). All MS/MS data were analyzed using Mascot 2.4 (Matrix Science, London, UK). MS data were searched against a decoyed database from UniProt (organism:7227, date:20131216) concatenated with an in-house build contaminant database. Precursor ion mass tolerance was set to 10 ppm and the fragment ion mass tolerance was set to 0.6 Da. The following search parameters were used: trypsin digestion (two missed cleavages

allowed), fixed modifications of carbamidomethyl labelled cysteine, and as variable modification oxidation of methionine, transformation of N-terminal glutamine to pyroglutamic acid (Gln→pyro-Glu), and deamidation of asparagine and glutamine. Data was filtered using Scaffold (v4.10.0) with the following settings: Peptide FDR: 0.1%, Protein FDR: 0.8%, and a minimum of two peptides per protein. Protein identified in less than 3 replicates were discarded using a python script (python 3.7). Protein interaction probabilities were calculated using SAINTExpress [88] through the CrapOme webtool (v2.0) [89]). For fold change calculation and interaction probability calculation replicates were averaged and ten virtual controls were processed for background estimation.

## Cell culture

S2R+ cell culture, transfection, and generation of stable cell lines was done essentially as previously described [83]. For time lapse imaging, cells were plated in 35mm glass-bottom dishes (Greiner Bio-One GmbH). Expression of Separase activity sensors was induced by addition of 250 μM CuSO4 to the culture medium and cells were imaged 48 hours later.

## Microscopy

Whole mount and squash preparations of testes were made as described previously [50]. DNA was stained using Hoechst 33258. Immunolabeling of UNO was done with affinity-purified rabbit polyclonal antibodies (rb34) against a C-terminal UNO peptide (MERETHRRWSAVPPEGNPV; Moravian Biotechnology, Brno, Czech Republic) at 1:1000. For Lamin immunolabeling we used the mouse monoclonal antibody ADL67.10 (Developmental Studies Hybridoma Bank, University of Iowa, Iowa City, IA, USA). Secondary antibodies used were Alexa 488 or Alexa 568 conjugated goat anti-rabbit IgG (Invitrogen A11008 and A11011) and Alexa 568 conjugated goat anti-mouse IgG (Invitrogen A11004) both at 1:600.

Fluorescence in situ hybridization (FISH) to testes squash preparations with a red fluorescent probe targeting chr X and a green fluorescent probe targeting chr Y was done as described [48]. For each genotype at least two slides were prepared and analyzed; on each slide testes dissected from 5–6 males were pooled before squash preparation.

In general, image stacks of the apical testis tip regions, as well as of cysts with early round spermatids, were acquired with a 40×/1.4 oil immersion objective on a Zeiss Cell Observer HS. Spacing between focal planes was 280 nm. For high resolution analyses, we acquired images stacks using a 100×/1.4 oil immersion objective on the same set up, with 240 nm spacing between focal planes. All images of fixed preparations (whole mounts and squashes) displayed in the figures represent maximum intensity projections.

For determination of the DNA content in early spermatid nuclei, we used testes squash preparations stained with Hoechst 33258. For each genotype at least two slides were prepared and analyzed; on each slide testes dissected from 5–6 males were pooled before squash preparation. For analysis we used a procedure described before [50] with the following modifications. Image stacks of cysts with round spermatids were acquired. Maximum intensity projections were analyzed using Fiji [90]. Regions of interest (ROIs) containing round spermatid cysts were defined manually using the freehand selection tool. The "reset histogram" function was used to normalize the histogram to the range displayed within the ROI. "Auto local threshold" was applied using the parameters method = Mean and radius = 15 pixels. The "watershed function" was used to split touching nuclei. "Analyze Particles" was used with the parameters size = 1–25 μm and circularity = 0.70–1.00. The resulting selection of spermatid nuclei was visually inspected and selected cyst cell nuclei were deleted. In the rare cases where more than 10% of the nuclei were not automatically selected, the image was deemed of poor

quality and discarded. Integrated intensity of the DNA signal was measured for all the retained spermatid nuclei. To facilitate analyses with the described procedure, a set of ImageJ macros as well as python scripts for result processing and visualization were generated and made available from the github repository (BioJoe/AHC-assay-by-DNA-content).

To match a *UASt-uno-EGFP* transgene insertion and a *UASt-uno^nc^-EGFP* transgene insertion with comparable expression levels, we acquired image stacks of the EGFP signals displayed in the apical testis tip region in whole mount preparations after crossing in the *bamP-GAL4-VP16* driver. Multiple transgene insertions were analyzed. Three distinct tip regions were imaged for each insertion followed by determination of the average expression level for each insertion. For signal quantification, a rectangular ROI (152 x 112 μm) was applied to the maximum intensity projections.

For the comparison of the weak UNO-EGFP dot signals in autosomal territories of spermatocytes with and without *tef* function, we imaged 8 and 9 cysts, respectively. In each cyst, three randomly chosen nuclei were visually inspected and the number of detected autosomal UNO-EGFP dot signals was scored.

To distinguish CG32117-EGFP signals clearly from autofluorescent background in the green channel, testes dissected from males carrying *gCG32117-EGFP* (III.1) and from males carrying *gitdTomato2x-Cenp-C* (III.3) were pooled for fixation, DNA staining and mounting. During microscopic analysis, control testes were identified based on the red fluorescent centromere signals. The same control flies and testes pool strategy were also used for the analysis of CG32117-EGFP signals resulting from *bamP-GAL4-VP16*-driven *UASt-CG32117-EGFP* expression.

Time lapse imaging of progression through meiosis was performed as recently described [56]. In brief, testes from pupal or young adult males were dissected in Schneider's *Drosophila* Medium (Invitrogen, #21720) supplemented with 10% fetal bovine serum (Invitrogen) and 1% penicillin/streptomycin (Invitrogen, #15140). The dissected testes were transferred into 40 μl of medium in a 35 mm glass bottom dish (MatTek Corporation, #P35G-1.5-14-C) and opened with fine tungsten needles to release the cysts. To reduce sample movements, 15 μl of 1% w/v methylcellulose (Sigma, #M0387) was added. A wet filter paper was placed inside along the dish wall before sealing the lid with parafilm. Imaging was performed at 25°C in a room with temperature control using a spinning disc confocal microscope (VisiScope with a Yokogawa CSU-X1 unit combined with an Olympus IX83 inverted stand and a Photometrics evolve EM 512 EMCCD camera, equipped for red/green dual channel fluorescence observation; Visitron systems, Puchheim, Germany) using a 60×/1.42 oil immersion objective. Image stacks with 25–39 focal planes spaced by 500 nm were acquired with a time interval of 45–50 seconds. Imaris software (Bitplane) was used to track centromere signals and for production of mp4 files from maximum intensity projections, as well as for exporting still frames, which were assembled using Photoshop (Adobe).

Signal intensity of UNO-EGFP in the dot associated with the sex chromosome bivalent was quantified using Imaris software (Bitplane). Regions of interest containing individual spermatocytes undergoing meiosis I were cropped out. An isosurface enclosing the UNO-EGFP dot was generated in the green channel and automatically tracked over time, followed by a manual correction of tracks if necessary. The integrated EGFP signal intensity within the isosurface was exported together with the time values. The ten time points with maximal signals were identified and the average intensity during these ten time points was set to 100 arbitrary units (a.u.). The time point just before the onset of NEBD was assigned as t = 0. In addition, we quantified the signal intensity of UNO-EGFP in the dot associated with the sex chromosome bivalent in a slightly different manner for the curves presented in S7 Fig. In this case, all the imaged spermatocytes of a cyst were analyzed simultaneously also by generating and

tracking isosurfaces over time. However, intensity values were not normalized, and the curves for the different spermatocytes, which progress somewhat asynchronously through M I, were not aligned temporally.

Time lapse imaging of stably transfected S2R+ cells was performed at 25˚C also with the spinning disc confocal microscope (see above). The 40×/1.3 oil immersion objective was used for imaging cells expressing the separase sensor proteins with the Rad21-derived linker region and the 60×/1.42 oil immersion objective was used for imaging the versions with UNO variants instead of the Rad21 linker region. Image stacks with 30 focal planes and 500 nm spacing were acquired at 4 minute time intervals. Imaris software (Bitplane) was used to crop movies and export still frames, which were assembled using Photoshop (Adobe).

## Supporting information

**S1 Fig. Interactomes of MNM-EGFP and SNM-EGFP. (A, B)** Scatter plots display the fold change of spectral counts (in comparison to control) as well as the probability of interaction for proteins co-purified with the baits. MNM-EGFP and SNM-EGFP were used as bait proteins, EGFP as control. Mass spectrometry was used for the identification of proteins co-purified by affinity purification from testis extracts with anti-GFP. Interaction probability was calculated by the SAINTexpress algorithm. **(A)** MNM-EGFP interactome. **(B)** SNM-EGFP interactome. The values obtained for the known AHC proteins and for UNO are highlighted (boxes). To avoid overcrowding of the SNM interactome plot, only preys with a fold change higher than five were annotated.
(PDF)

**S2 Fig. Alignment of Uno protein sequence.**
(PDF)

**S3 Fig. Alignment of CG32117 protein sequences.**
(PDF)

**S4 Fig. DNA content in early postmeiotic spermatid nuclei.** Squash preparations of testes from the indicated genotypes were labeled with a DNA stain, and cysts with early postmeiotic spermatids were imaged. DNA signal intensity in each spermatid nucleus was quantified. The average of all the individual nuclear DNA content values obtained for a given cyst was used for normalization of these values. The normalized values obtained for the nuclei within a given cyst are plotted in the same color with different cysts alternating between black and red from left to right. The standard deviation of the nuclear DNA content values within a given cyst was used for ordering the different cysts of the same genotype, with standard deviations increasing from left to right. The average standard deviation for all analyzed cysts of the same genotype is displayed along with the number of analyzed cysts.
(TIF)

**S5 Fig. *uno* is not required for faithful chromosome segregation in M II. (A,B)** Chromosome segregation during meiosis was analyzed by time-lapse imaging with *uno* null mutant spermatocytes (*uno*^cc1^/ *Df(2R)Exel7094*) expressing Cenp-A/Cid-EGFP and His2Av-mRFP. **(A)** Because of random segregation of eight univalents during M I, pairs of secondary daughter spermatocytes can arise in *uno* null mutants with a centromere distribution that is not 4:4, as after regular separation of four bivalents during normal M I. In case of the two *uno* null mutant spermatocytes documented, centromeres were segregated 5:3 (cell 1) and 6:2 (cell 2) during M I. The corresponding numbers of Cenp-A/Cid-EGFP dots are clearly detectable in the still frames selected at the start of M II (NEBD II). After random segregation during M I, however,

*uno* null mutants segregate sister centromeres regularly during M II, as illustrated with still frames selected from anaphase II with centromeres marked by small spheres. Note that spinning disc confocal microscopy of Cenp-A/Cid-EGFP cannot resolve the two sister centromeres before bi-orientation [56]. **(B)** Progression of a *uno* null mutant secondary spermatocyte through M II. The indicated times (min:sec) are given relative to onset of NEBD II. From left to right, characteristic still frames from the stages interkinesis (- 42:45), early (06:45) and late (15:45) prometaphase, metaphase (23:15), early (27:45), mid (30:00) and late (33:00) anaphase document that the three univalents present before onset of M II bi-orient normally into a metaphase plate followed by separation of sister centromeres and regular segregation to opposite poles. Scale bar = 2 μm.
(TIF)

**S6 Fig. Characterization of CG32117. (A)** *CG32117* transgenes and mutant alleles. The genomic region containing *CG32117* is shown schematically. Untranslated regions (white boxes) and coding regions (black boxes) are indicated. The region present in the *g-CG32117-EGFP* transgene is marked with a green bar. A red bar indicates the regions deleted by the four intragenic out-of-frame deletions induced by CRISPR/cas (cc1, cc2, cc3, and cc4). The precise breakpoints in these four alleles are distinct, although not apparent at the displayed scale. **(B)** *CG32117* expression pattern. Whole mount preparations of testes isolated from either control males (middle column, third and fourth panel from top) and *g-CG32117-EGFP* males (all other panels) were fixed and stained for DNA. The low magnification view (left column) reveals that *g-CG32117-EGFP* is expressed in cysts with fully elongated spermatids. CG32117-EGFP signals in the tail region of such spermatids are shown at higher magnification in the middle columns (top two panels), and the comparison with control (bottom two panels) demonstrates that these signals are above the autofluorescent background. CG32117-EGFP signals in the head region of fully elongated spermatids are shown at higher magnification as well (right column). Scale bars = 200 μm (left column), 20 μm (middle column), and 10 μm (right column). **(C)** Forced premature CG32117-EGFP expression in early spermatocytes. A *UASt-CG32117-EGFP* transgene was expressed with *bam-GAL4-VP16*. In the top row, maximum intensity projections of apical testis regions from whole mount preparations stained for DNA are shown. The comparison with control preparations (right) indicates that CG32117-EGFP (left) is clearly detectable but only transiently in early spermatocytes. Higher magnification views from single optical sections with early spermatocytes (middle and bottom rows) illustrate the subcellular localization of CG32117-EGFP. Apart from strong nuclear dots that do not appear to be chromatin-associated (middle row), a weak diffuse signal throughout the nucleus is evident with display settings enhancing signal intensity in the green channel (bottom row). Scale bars in C = 20 μm (top row) and 10 μm (middle and bottom rows).
(TIF)

**S7 Fig. Dynamics of bivalent splitting and disappearance of AHC proteins during M I. (A, B)** Time-lapse imaging with spermatocytes expressing His2Av-mRFP and UNO-EGFP. **(A)** Still frames illustrate progression through M I in spermatocytes expressing UNO-EGFP in an *uno*+ background. Time (hours:minutes:seconds) after the last time point before onset of NEBD I is indicated. **(B)** Quantification of UNO-EGFP signal intensity in the XY-associated dot during progression through M I in spermatocytes expressing UNO-EGFP in an *uno*+ background (left panel) or instead of endogenous *uno*+ (right panel). In both panels, several spermatocytes (6 and 8 cells, respectively) were analyzed that were present in the same cyst and therefore divided with high but not perfect synchrony. An arbitrary time point when all spermatocytes were still in metaphase was set to zero. Arrows above the curves indicate the time point when autosomes started to move rapidly and steadily towards opposite poles, i.e., when

the final linkages between homologs in the autosomal bivalents were severed presumably. **(C, D)** Time-lapse imaging with spermatocytes expressing His2Av-mRFP, Cenp-A/Cid-EGFP and MNM-EGFP. The metaphase to anaphase I transition in a representative spermatocyte is documented. An arbitrary time point in late metaphase I was set to zero. **(C)** Still frames at t = 3:10, 6:40 and 7:20 represent late metaphase before the onset of increased bivalent stretching, an early anaphase stage where autosomes but not the sex chromosomes have started to move apart rapidly, and a later anaphase stage where also the sex chromosomes have started to move apart. Centromeres are marked by colored dots (chr XY red, chr 2 yellow, chr 3 blue, chr 4 green; see [56] for explanation of bivalent identification), the MNM-EGFP dot at the chr XY pairing site with an arrowhead. **(D)** MNM-EGFP intensity in the XY-associated dot and Cid-EGFP intensity in a selected centromere (to monitor bleaching), as well as the distances between the two centromeres of a bivalent were quantified. The dotted vertical lines mark the time point when increased bivalent stretching started (left line) and when the final linkages between the homologs in autosomal bivalents were severed apparently (right line), as suggested by the subsequent rapid and steady poleward autosomal centromere movements. Note the characteristic delayed splitting of the sex chromosome bivalent. Scale bars = 3 μm. (TIF)

**S8 Fig. Testis mass isolation. (A,B)** Light microscopic images for illustration of the purity of the final testes fraction obtained by mass isolation from late larvae and early pupae. As indicated in Materials and Methods, final testis purity depends critically on the level of the GFP signal in testis and on parameter settings chosen for fluorescent particle sorting. In the optimal case, the fraction of contaminating bits of tissue other than testis is around 3%, as evident from (A). However, testes have residual fat body attachments to variable degrees, as evident from the examples shown in (B), with fat body remnants displaying a characteristic brighter appearance. Scale bars = 1 mm (A) and 100 μm (B). (TIF)

**S1 Movie. Random segregation of univalents during M I in uno mutant spermatocyte.** Progression through M I was analyzed by time-lapse imaging (z stacks of 45 sections with 500 nm spacing at 45 sec intervals using 60x) with *uno* mutants (*uno*cc1/ *Df(2R)Exel7094*) expressing Cenp-A/Cid-EGFP (green) and His2Av-mRFP (grey values). Centromeres were tracked and a Gaussian filter was applied to the His2Av-mRFP channel. A maximum intensity projection is shown. (MP4)

**S2 Movie. Localization of UNO-EGFP during M I.** Spermatocytes expressing His2Av-mRFP (magenta) and UNO-EGFP (green) instead of endogenous UNO protein (*bam>uno-EGFP* in *uno*cc1 homozygotes) were analyzed by time-lapse imaging (z stacks of 37 sections with 500 nm spacing at 49 sec intervals using 60x). A maximum intensity projection of a representative spermatocyte is shown. (MP4)

**S3 Movie. Localization of UNO-EGFP on autosomal bivalents during M I.** Spermatocytes expressing His2Av-mRFP (magenta) and UNO-EGFP (green) instead of endogenous UNO protein (*bam>uno-EGFP* in *uno*cc1 homozygotes) were analyzed by time-lapse imaging (z stacks of 37 sections with 500 nm spacing at 49 sec intervals using 60x). A maximum intensity projection of a spermatocyte with well visible UNO-EGFP dots on autosomal bivalents during early M I is shown three times. During the first repeat, display settings in the green channel were adjusted to avoid saturation of the XY-associated UNO-EGFP dot. During the second repeat, display settings in the green channel were adjusted to reveal the weak UNO-EGFP dots

on autosomal bivalents. The third repeat presents exclusively the enhanced green signals in grey values.
(MP4)

**S4 Movie. Localization of UNO-EGFP during M I in a uno+ background.** Spermatocytes expressing His2Av-mRFP (magenta) and UNO-EGFP (green) (*bam>uno-EGFP*) were analyzed by time-lapse imaging (z stacks of 30 sections with 762 nm spacing at 45 sec intervals using 60x). A maximum intensity projection of a representative spermatocyte is shown.
(MP4)

**S5 Movie. MNM-EGFP and bivalent separation during the metaphase to anaphase I transition.** Spermatocytes expressing His2Av-mRFP (magenta) and Cenp-A/Cid-EGFP (green) and MNM-EGFP (green) were analyzed by time-lapse imaging (z stacks of 32 sections with 300 nm spacing at 10 sec intervals using 60x). Centromeres were tracked and marked with colored dots (chr XY red, chr 2 yellow, chr 3 blue, chr 4 green). The MNM-EGFP dot at the chr XY pairing site in the center was not labeled. A maximum intensity projection of a representative M I spermatocyte progressing from metaphase into anaphase is shown.
(MP4)

**S6 Movie. Inhibition of UNO-EGFP disappearance during M I after separase depletion.** Spermatocytes expressing His2Av-mRFP (magenta) and UNO-EGFP (green), as well as a $thr^{RNAi}$ transgene to eliminate separase activity (*bam>uno-EGFP* and $thr^{RNAi}$) were analyzed by time-lapse imaging (z stacks of 30 sections with 500 nm spacing at 45 sec intervals using 60x). A maximum intensity projection of a representative spermatocyte is shown.
(MP4)

**S7 Movie. Progression through M I in the presence of UNOnc-EGFP (cell 1).** Spermatocytes expressing His2Av-mRFP (magenta) and $UNO^{nc}$-EGFP (green) instead of endogenous UNO protein (*bam>uno-EGFP* in $uno^{cc1}$ homozygotes) were analyzed by time-lapse imaging (z stacks of 45 sections with 500 nm spacing at 45 sec intervals using 60x). A maximum intensity projection of a spermatocyte close to the coverslip during M I is shown after applying a drift correction in three re-iterations. During the first repeat, display settings in the green channel were adjusted to avoid saturation of the XY-associated UNO-EGFP dot. During the second repeat, display settings in the green channel were adjusted to reveal the weak UNO-EGFP dots on autosomal bivalents. The third repeat presents exclusively the enhanced green signals in grey values.
(MP4)

**S8 Movie. Progression through M I in the presence of UNOnc-EGFP (cell 2).** A spermatocytes with well-visible autosomal $UNO^{nc}$-EGFP after imaging and processing as described in the legend for S6 Movie is presented.
(MP4)

**S9 Movie. Progression through M I in the presence of both UNOnc-EGFP and UNO.** Spermatocytes expressing His2Av-mRFP (magenta) and $UNO^{nc}$-EGFP (green) in a $uno^+$ background (*bam>uno-EGFP* in $uno^+$) were analyzed by time-lapse imaging (z stacks of 30 sections with 782 nm spacing at 45 sec intervals using 60X). A maximum intensity projection of a spermatocyte after drift correction is shown in three re-iterations. During the first repeat, display settings in the green channel were adjusted to avoid saturation of the XY-associated UNO-EGFP dot. During the second repeat, display settings in the green channel were adjusted to reveal the weak UNO-EGFP dots on autosomal bivalents. The third repeat presents exclusively the enhanced green signals in grey values.
(MP4)

**S10 Movie. Exit from M I in the presence of both UNOnc-EGFP and UNO.** Spermatocytes expressing His2Av-mRFP (magenta) and UNO<sup>nc</sup>-EGFP (green) in a *uno*<sup>+</sup> background (*bam>uno-EGFP* in *uno*<sup>+</sup>) were analyzed by time-lapse imaging (z stacks of 30 sections with 782 nm spacing at 45 sec intervals using 60X). A maximum intensity projection of a spermatocyte progressing through metaphase and anaphase of M I is shown in three re-iterations. During the first repeat, display settings in the green channel were adjusted to avoid saturation of the XY-associated UNO-EGFP dot. During the second repeat, display settings in the green channel were adjusted to reveal the weak UNO-EGFP dots on autosomal bivalents. The third repeat presents exclusively the enhanced green signals in grey values.
(MP4)

**S1 Table. Proteins co-purified with MNM-EGFP, SNM-EGFP and EGFP.** Mass spectrometry was used for identification of co-purified proteins. Spectral counts of the detected peptides are given for each of the identified proteins. Moreover, a fold change of spectral counts in comparison with control (EGFP) and an interaction probability as calculated by SAINTExpress is indicated as well.
(XLSX)

**S2 Table. Description of the analyzed genotypes.**
(XLSX)

**S3 Table. Oligonucleotide sequences.**
(XLSX)

**S4 Table. Source data.**
(XLSX)

## Acknowledgments

We thank B. Roschitzki (Functional Genomics Center Zurich/UZH/ETHZ) and J. Matos (ETHZ, Institute of Biochemistry) for help and advice with mass spectrometric analysis, as well as S. Moser and H. Yamada for technical support.

## Author Contributions

**Conceptualization:** Joe Weber, Christian F. Lehner.

**Data curation:** Joe Weber, Christian F. Lehner.

**Formal analysis:** Joe Weber, Zeynep Kabakci, Soumya Chaurasia, Christian F. Lehner.

**Funding acquisition:** Joe Weber, Christian F. Lehner.

**Investigation:** Joe Weber, Zeynep Kabakci, Soumya Chaurasia.

**Methodology:** Joe Weber, Erich Brunner, Christian F. Lehner.

**Project administration:** Christian F. Lehner.

**Resources:** Erich Brunner, Christian F. Lehner.

**Software:** Joe Weber.

**Supervision:** Christian F. Lehner.

**Validation:** Joe Weber, Christian F. Lehner.

**Visualization:** Joe Weber, Christian F. Lehner.

**Writing – original draft:** Joe Weber, Christian F. Lehner.

**Writing – review & editing:** Joe Weber, Zeynep Kabakci, Erich Brunner, Christian F. Lehner.

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
