## [Decision Letter · Decision Letter 0]

25 May 2020

Dear Dr Lehner,

Thank you very much for submitting your very interesting Article entitled 'Chromosome separation during Drosophila male meiosis I requires separase-mediated cleavage of the homolog conjunction protein UNO' to PLOS Genetics.  Your manuscript was fully evaluated at the editorial level and by independent peer reviewers. All three reviewers are enthusiastic about the quality and the relevance of your work. We would be very happy to publish your manuscript in PLOS Genetics. However, the reviewers noted some minor points (such as some quantifications), which can be addressed easily before publication. I strongly encourage you to make these modifications, which can only improve your manuscript. Your revisions should address the specific points made by each reviewer.

[LINK]

Yours sincerely,

Jean-René Huynh

Associate Editor

PLOS Genetics

Gregory P. Copenhaver

Editor-in-Chief

PLOS Genetics

Reviewer's Responses to Questions

**Comments to the Authors:**

Reviewer #1: review uploaded as an attachment

Reviewer #2: Meiosis in Drosophila males employs a different system to segregate chromosomes during meiosis. A novel set or proteins, not used in the more conventional female meiosis, connect the homologs. These are referred to as the alternative homolog conjunction (AHC) system. All the components of this system are not known and many details of how it works are also not known. This paper reports on the identification of a new AHC member, UNO, using a recently developed method to mass isolate testis material from larval males and then identify proteins that interact with MNM or SNM.

The author show quite convincingly that UNO is a new protein required for conjunction and segregation of homologs in male meiosis. There is nice data showing colocalization with MNM, SNM and some interdependencies. Beautiful live imaging shows loss of UNO with entry into anaphase I. Not only is cytology used to show loss of UNO depends on Separase, but an elegant in vivo cleavage assay is used to show that UNO is cleaved by Separase. Like the other members, UNO behaves like a protein holding homologs together. If there is any criticism, it would be this paper has identified a new member of the AHC but not revealed significant insights into how the AHC complex works. However, I think this paper is significant and of interest to the meiosis community because these results support the hypothesis that the AHC ensure meiosis I segregation using a Separase based mechanism. This is significant. That there are interesting parallels (and lack of parallels as well) between UNO and the Kleisin family is interesting. UNO could be an important discovery in understanding how the AHC system works.

Thus, I have no major concerns and only the following minor ones. They are mostly related to presentation of the data in the figures and the absence of some sample sizes in several experiments.

1) The most significant criticism is that the data in a few locations lacks quantitation. For example, in Figure 2B, it is difficult to appreciate how often uno mutants have defects in S5 and promet I. The spermatid images look dramatic but how often do they occur? The prometa I images do not look as bad in the mutant. Can the author conclude the defects get progressively worse? And that Uno is not necessary for pairing? Reporting the numbers of chromosomes at each stage would be helpful.

2) Figure 2F is hard to interpret without a wild-type image. The univalent is obvious, as is the aberrant segregation at anaphase, but it is unclear if the other chromosomes are misbehaving in Prometa I. How rapidly do the bivalents dissociate?

3) What is the sample size in 2E and 2G? It is also a little surprising there is no genetic measures of nondisjunction.

4) Figure 3 could use more single channel images, especially for UNO in panel C. Can the authors also confirm whether the autosomal foci are never seen with antibody? (suggested in line 361-363). Can the authors give an estimate of how often the autosomal signals are observed?

5) In Line 201: can the authors provide any more information on why they picked CG8712 for detailed analysis?

6) In Figure 4, it seems that UNO most strongly affects MNM, and SNM most strongly affects UNO. Therefore, can the authors conclude any sort of hierarchy, such as SNM – UNO – MNM? (MNM depends on UNO and UNO depends on SNM). Granted, it is not clean and there are partial dependencies between all members as expected for a complex. It is just surprising that if it is a complex, that UNO localizes quite strongly in a mnm mutant.

7) Is the strong UNO on the X affected by tef? It looks that way in Fig 4C.

8) It would help to have single channel images of UNO in Figure 4C, particularly for the images showing the single nucleus and lack of autosomal signals (as is done for mnm-). This should include a wild-type single nucleus image to compare to the mutants in Fig 4C.

9) In Figure 4C, the UNO signal on the X-Y pair looks dimmer in the tef mutant.

10) Why was the cleavage assay was done in S2 cells. Does it not work in testis?

11) Figure 7D shows segregating autosomes with a slight delay. How often is this observed? This panel is presented but there is no data on how often this was repeated or the delays observed. When the autosomes segregate, do they have autosomal UNO?

12) Do other AHC proteins have separase cleavage sites?

13) The paragraph starting in line 619 appears to contradict itself. The first line suggests that there is a relationship between UNO and kleisins. Line 630 suggest the data argues against a relationship. What do the authors favor? The idea that UNO has at least a kleisin-like role is attractive given the results of this paper. This is particularly important and suggests that, despite the lack of SMC1 and SMC3, that SNM/MNM/UNO form a ring complex that holds homologs together. Or can the authors suggest another mechanism for how Separase cleavage allows the homologs to separate?

Reviewer #3: The manuscript by Weber et al is addressing how bivalents are separated in drosophila male meiosis. In drosophila, homologs do not recombine, and no crossover formation or formation of a synaptonemal complex is observed, hence it is mysterious how homologs are held together until metaphase I to correctly align at the bipolar spindle and segregate correctly into two daughter cells.

Using a novel approach to obtain large quantities of testes for proteomics, the authors were able to identify a protein they call Uno as an interaction partner of Mnm and Snm, two proteins that were previously shown to be required for correct segregation of homologs. Through a series of impressive live imaging experiments, genetic invalidation and localization studies they show that Uno is required for correct segregation of homologs, localizes to chromosomes with a very prominent localization to the sex chromosomes, and disappears at anaphase I onset. Using a Separase cleavage sensor construct and a mutant approach they convincingly show that Uno is cleaved by Separase at anaphase I onset. In agreement with a role only for homolog segregation, Uno is not required for the second meiotic division.

In my opinion this is a very interesting manuscript, contributing in a significant manner to our understanding of homolog pairing and segregation without recombination, in this species. The experiments are convincing, and support the conclusions. (The live imaging movies are impressive!) In my opinion, this manuscript can be published in PloS Genetics and I have only the following minor comments:

1) Can the authors show a control in Figure 2F? For somebody not working on drosophila this would be helpful.

2) Figure 4C: It would be nice if the authors could show all stages (prometaphase I and S5) for all genetic backgrounds and the control

3) Concerning the Separase cleavage site I was wondering whether the authors have been able to identify a Separase docking site downstream of the cleavage site (LPE, Rosen et al. Nat Comm 2019). Are there Separase docking sites in other Separase substrates in drosophila?

**Have all data underlying the figures and results presented in the manuscript been provided?**

Reviewer #1: Yes

Reviewer #2: Yes

Reviewer #3: Yes

PLOS authors have the option to publish the peer review history of their article (what does this mean?). If published, this will include your full peer review and any attached files.

Reviewer #1: Yes: Bruce D. McKee

Reviewer #2: No

Reviewer #3: No

---

## [Editor Report · Decision Letter 1]

10 Jun 2020

Dear Dr Lehner,

We are pleased to inform you that your manuscript entitled "Chromosome separation during Drosophila male meiosis I requires separase-mediated cleavage of the homolog conjunction protein UNO" has been editorially accepted for publication in PLOS Genetics. Congratulations!

Yours sincerely,

Jean-René Huynh

Associate Editor

PLOS Genetics

Gregory P. Copenhaver

Editor-in-Chief

PLOS Genetics

Comments from the reviewers (if applicable):

**Data Deposition**

http://datadryad.org/submit?journalID=pgenetics&manu=PGENETICS-D-20-00631R1

**Press Queries**

---

## [Editor Report · Acceptance letter]

16 Jul 2020

PGENETICS-D-20-00631R1 

Chromosome separation during Drosophila male meiosis I requires separase-mediated cleavage of the homolog conjunction protein UNO 

Dear Dr Lehner, 

We are pleased to inform you that your manuscript entitled "Chromosome separation during Drosophila male meiosis I requires separase-mediated cleavage of the homolog conjunction protein UNO" has been formally accepted for publication in PLOS Genetics! Your manuscript is now with our production department and you will be notified of the publication date in due course.

With kind regards,

Matt Lyles

PLOS Genetics

On behalf of:
